



*Manuscript type: Development and Technical Paper*

# Dynamic ecosystem assembly and escaping the "fire-trap" in the tropics: Insights from FATES_15.0.0

Jacquelyn K. Shuman[1,2], Rosie A. Fisher[3], Charles Koven[4], Ryan Knox[4], Lara Kueppers[4,5] and Chonggang Xu[6]

[1]Earth Science Division, NASA Ames Research Center, P.O. Box 1, M/S N245-4, Moffett Field, CA, 94035-0001, USA
[2]Climate and Global Dynamics, National Center for Atmospheric Research, P.O. Box 3000, Boulder, CO, 80307-3000, USA
[3]CICERO Center for International Climate Research, Oslo, 87350, Norway
[4]Climate and Ecosystem Sciences Division, Lawrence Berkeley National Laboratory, 1 Cyclotron Road, Berkeley, CA, 94720, USA
[5]Energy and Resources Group, University of California, Berkeley, 345 Giannini Hall #3050, Berkeley, CA 94720, USA
[6]Earth and Environmental Sciences Division, Los Alamos National Laboratory, P.O. Box 1663, Los Alamos, NM, 87545, USA

*Correspondence to*: Jacquelyn K. Shuman (Jacquelyn.k.shuman@nasa.gov)

**Abstract.**

Fire is a fundamental part of the Earth system, with impacts on vegetation structure, biomass and community composition, the latter mediated in part via key fire-tolerance traits, such as bark thickness. Due to anthropogenic climate change and land use pressure, fire regimes are changing across the world, and fire risk has already increased across much of the tropics. Projecting the impacts of these changes at global scales requires that we capture the selective force of fire on vegetation distribution through vegetation functional traits and size structure. We have adapted the fire-behavior and effects module, SPITFIRE, for use with the Functionally Assembled Terrestrial Ecosystem Simulator (FATES), a size-structured vegetation demographic model. We test how climate, fire regime and fire-tolerance plant traits interact to determine the biogeography of tropical forests and grasslands. We assign different fire-tolerance strategies based on crown, leaf and bark characteristics, which are key observed fire-tolerance traits across woody plants. For these simulations, three types of vegetation compete for resources: a fire-vulnerable tree with thin bark, a vulnerable deep crown and fire-intolerant foliage; a fire-tolerant tree with thick bark, a thin crown and fire-tolerant foliage; and a fire-promoting C4 grass. We explore the model sensitivity to a critical parameter governing fuel moisture, and show that drier fuels promote increased burning, an expansion of area for grass and fire-tolerant trees and a reduction of area for fire-vulnerable trees. This conversion to lower biomass or grass areas with increased fuel drying results in increased fire burned area and its effects, which could feedback to local climate variables. Simulated size-based fire mortality for trees less than 20 cm in diameter and those with fire-vulnerable traits is higher than that for larger and/or fire-tolerant trees, in agreement with observations. Fire-disturbed forests demonstrate reasonable productivity and capture observed patterns of aboveground biomass in areas dominated by natural vegetation for the recent historical period,





but have a large bias in less disturbed areas. Though the model predicts a greater extent of burned fraction than observed in
areas with grass dominance, the resulting biogeography of fire-tolerant, thick-bark trees and fire-vulnerable, thin-bark trees
corresponds to observations across the tropics. In areas with more than 2500 mm of precipitation, simulated fire frequency and
burned area are low, with fire intensities below 150 kW m$^{-1}$, consistent with observed understory fire behavior across the
Amazon. Areas drier than this demonstrate fire intensities consistent with those measured in savannas and grasslands, with
high values up to 4000 kW m$^{-1}$. The results support a positive grass-fire feedback across the region, and suggest that forests
which have existed without frequent burning may be vulnerable at higher fire intensities, which is of greater concern under
intensifying climate and land use pressures. The ability of FATES to capture the connection between fire disturbance and plant
fire-tolerance strategies in determining biogeography provides a useful tool for assessing the vulnerability and resilience of
these critical carbon storage areas under changing conditions across the tropics.

**1 Introduction**

Fire is a fundamental component of the Earth system, with a diversity of global fire regimes playing a role in determining
vegetation distribution, composition and structure, and carbon storage (Pausas and Keeley, 2014; McLauchlan et al., 2020).
Recent decades show changing fire conditions with increases in fire season length (Jolly et al., 2015; Jones et al., 2022), driven
largely by hotter and drier conditions (Jain et al., 2022). It is expected that these changes will continue with rising greenhouse
gas emissions, leading to further elevation in fire risk (Touma et al., 2021). The projection of fire under changing climate and
CO$_2$ conditions is challenging (Hantson et al., 2020) due to the many drivers of fire (climate, fuel properties, land management,
anthropogenic activities) that are simultaneously evolving. For tropical forests in particular (Cochrane, 2003; Cochrane et al.,
1999; Nepstad et al., 2008; Nobre et al., 2016), these increasing drivers represent an emergence of conditions that have no
observable analogue in the present day. Thus, while purely data-driven approaches can help inform how combinations of
drivers affect fire behaviour (Haas et al., 2022), we must rely on (and improve) process-based models to effectively project
these rapidly changing fire regimes and their effects.

In many land surface models - the terrestrial components of Earth system models (Blyth et al., 2021) - fire is represented as a
function of fuel availability and dryness, climate conditions, and human activity (Rabin et al., 2017). Most land surface models,
however, do not resolve the size distribution of plants nor variation in fire-tolerance traits, omissions with potentially important
implications. First, smaller trees and grasses are more prone to direct consumption by fire while larger trees can place more of
their branches and leaves clear of flames that are produced by surface fires. Second, tree mortality from fire of a given intensity
and duration is also known to be a strong function of bark thickness (Hoffmann et al., 2003; Hoffmann and Solbrig, 2003;
Hoffmann et al., 2012). Bark thickness varies as a function of tree size as well as tree type, and both size and bark investment
determine the survival of trees or stems during fire (Balch et al., 2008; Hoffmann and Solbrig, 2003; Hoffmann et al., 2012;
Pellegrini et al., 2017). Size-dependent mortality gives rise to the concept of the 'fire trap', upon which many aspects of fire





ecology are thought to depend (Bond, 2008; Ryan and Williams, 2011; Hoffmann et al., 2020), including bimodal size distributions, and the selection for thick-barked tree species under frequent fire regimes (Pellegrini et al., 2017). The latter implies that in areas where fire is rare, the absence of selection for thick bark will mean that trees are more vulnerable to mortality under a given set of fire conditions. Thus, models that do not differentiate plant types based on their size and tolerance of fire (expressed via bark thickness and canopy characteristics), may not capture these dynamics.

Here we describe the implementation of the process-based fire module, SPITFIRE (Thonicke et al., 2010), into the vegetation demographic model, FATES (Fisher et al., 2015; Koven et al., 2020), and test its ability to capture vegetation-fire feedbacks and ecosystem composition across tropical South America. FATES (the Functionally Assembled Terrestrial Ecosystem Simulator) is one of a class of demographic models presently being implemented in Earth System Models (Fisher et al., 2018; Naudts et al., 2015; Chen et al., 2018; Haverd et al., 2018). It captures heterogeneity of plant size by tracking populations of co-occurring plants using a set of 'cohorts' that recruit, grow in size, and die through time on a discrete set of 'patches' which vary in age since disturbance and collectively track succession following canopy mortality events. The fire module 'SPITFIRE' (SPread and InTensity of FIRE) (Thonicke et al., 2010), which is already used in other land surface schemes (Rabin et al., 2017), and incorporates size-dependent mortality algorithms, is implemented within FATES, and modified to facilitate the interaction between the size and age structured vegetation. To test the influence of fire on ecosystem assembly, we use FATES to simulate the distribution of forest and grass under multiple fuel drying conditions, evaluating size-dependent mortality and associated fire behavior and effects. The results of these simulations provide insight into the extent to which fire feedbacks regulate ecosystem assembly.

## 2 Materials and Methods

### 2.1 The integrated vegetation-fire model FATES-SPITFIRE

FATES-SPITFIRE has been integrated into the land models of both the Community Earth System Model (CESM, (Danabasoglu et al., 2020)) and the Energy Exascale Earth System Model (E3SM, (Golaz et al., 2019)) (the Community and E3SM Land Models (CLM and ELM), respectively). This study used FATES within the CLM to develop the climate-fire-vegetation interactions and feedbacks at regional scale. Similar to many land surface models, the default CLM wildfire scheme does not consider size cohorts and evaluates fire impact on vegetation and the carbon cycle as a weighted fraction of the fractional coverage of vegetation within the grid cell with fire altering the biomass and area of each vegetation type. The use of FATES allows for the inclusion of size and age structured vegetation and consideration of differential size-dependent mortality and associated fire behavior and effects.



### 2.1.1 FATES

We use FATES (version: ctsm5.1.dev036-fates_api15.0.0_crown_scorch_damage with git hash version number ff1ae2c2-a3b92952) which has been described most recently by (Koven et al., 2020), based on initial description by (Fisher et al., 2015, 2010). Recent application of FATES include investigation of vegetation dynamics in Western US ecosystems in the presence of fire (Buotte et al., 2021). The model code used here for the non-fire elements of this version of FATES is consistent with that documented in (Koven et al., 2020).

### 2.1.2 SPITFIRE

The process-based fire behavior and effects module SPITFIRE (Spread and InTensity of FIRE; (Thonicke et al., 2010)) is implemented in multiple vegetation models (e.g. (Lasslop et al., 2014; Yue et al., 2014; Drüke et al., 2019)) with complete technical details for this implementation found in Supplementary Material Section 3. In FATES, the SPITFIRE module operates at a daily timestep and separately for each patch to allow for sub-grid representation of different litter pools and vegetation characteristics according to the FATES patch structure. SPITFIRE simulates fires through calculation of fire danger, ignition, behavior and effects for live and dead vegetation fuels. Here we review the structure of the SPITFIRE module, and introduce modifications specific to its implementation in FATES.

#### 2.1.2.1 Ignitions and fire danger

Within FATES-SPITFIRE, anthropogenic ignitions and natural lightning strikes are both potential sources of ignition. Lightning strikes are prescribed by a lightning forcing dataset used in (Li et al., 2013) derived from the NASA LIS/OTD Gridded Climatology (http://ghrc.msfc.nasa.gov), assuming that a percentage of these strikes reach the ground to result in lightning-driven potential ignitions ($I_{lightning}$) (strikes km$^{-2}$ day$^{-1}$). For this study the percentage of cloud-to-ground lightning strikes under favourable conditions for burning is set at 10% (Latham and Williams, 2001). In this study due to the focus on natural fire-vegetation feedbacks, anthropogenic ignitions ($I_{anthro}$) were not used and instead set to zero. When in use, anthropogenic ignitions (strikes km$^{-2}$ day$^{-1}$) are calculated according to (Li et al., 2012) with details included in Supplementary Material Section 3. Fire duration ($F_{dur}$) is calculated as a function of the fire danger index ($FDI$) with a maximum daily duration of 240 min (Thonicke et al., 2010). $FDI$, a representation of the effect of meteorological conditions on the likelihood of a fire, is computed daily by using the Nesterov Index ($NI$), which is a cumulative function of daily temperature ($T$) and dew point ($Dew$) that resets to zero when total precipitation exceeds 3.0 mm. See Supplementary Material Section 3 for further details.

$$NI = \sum T * (T - Dew) \tag{1}$$

$$FDI = 1 - e^{-a*NI} \tag{2}$$





where $a = 0.00037$ per (Venevsky et al., 2002).

**2.1.2.2 Characteristics of Fuel**

The rate of spread, fire intensity and fuel combustion are determined based on multiple fuel conditions: fuel loading ($w$, kg m$^{-2}$), bulk density ($BD$) (kg m$^{-3}$), surface area-to-volume ratio ($SAV_{fc}$) (cm$^{-1}$), moisture ($moist_{fc}$) (m$^3$ m$^{-3}$) and moisture of extinction ($moist_{ext, fc}$) (m$^3$ m$^{-3}$). Weighted averages across fuel classes ($fc$) are calculated for each of these variables. Total fuel load ($F_{patch}$) (kg m$^{-2}$) is the sum of the aboveground coarse woody debris ($CWD_{AG,fc}$), leaf litter ($l_{litter}$), and live grass biomass ($b_{l,grass}$). As in (Thonicke et al., 2010), fuels are separated into multiple classes. Dead woody fuels are grouped according to diameter ranges associated with a timelag that defines the time necessary for the loss of initial moisture to attain an equilibrium moisture content (NWCG, 2002) per the methods of (Rothermel, 1983; Fosberg, 1971). According to this relationship, these dead woody fuels are categorized by their diameter as 1-hr for fuels less than 0.6 cm, 10-hr for fuels between 0.6 and 2.5 cm, 100-hr for those between 2.5 and 7.6 cm, and 1000-hr for fuels greater than 7.6 cm (NWCG, 2002). Fine and woody fuels accumulate according to litterfall and size-differentiated mortality inputs produced by FATES and temperature- and moisture-sensitive litter decomposition within CLM (Lawrence et al., 2019). The rates of decomposition transfer for fuels were updated for the 1-hr, 10-hr and 100-hr fuels according to (Eaton and Lawrence, 2006), 1000-hr fuels per (Chambers et al., 2000), and dead leaves per (Thonicke et al., 2010) (Table 1). The impact of 1000-hour fuels on mean fuel properties is not considered in rate of spread or fire intensity equations, but they can be combusted during a fire.

Dead fuel moisture ($moist_{fc}$) is calculated as:

$$moist_{fc} = e^{-\propto_{fc} NI} \tag{3}$$

$$\propto_{fc} = \frac{SAV_{fc}}{drying\ ratio} \tag{4}$$

Live grass fuel moisture ($moist_{l,grass}$) is calculated as:

$$moist_{l,grass} = e^{-\propto_{1hr,fc} NI} \tag{5}$$

where $\propto_{fc}$, a user-defined parameter, indicates the rate of drying of the fuel classes. Lower $drying\ ratio$ values are associated with more rapid drying and lower relative moisture (Figure S1), which in turn impacts fuel combustion (Figure S2). The moisture of extinction, the moisture content (m$^3$ m$^{-3}$) at which fuel can no longer burn, is calculated as in (Peterson and Ryan, 1986):





$$moist_{ext,fc} = 0.524 - 0.066 \, log_{10}SAV_{fc} \tag{6}$$

Effective fuel moisture is then the ratio of fuel moisture $moist_{fc}$ to $moist_{ext,fc}$ and used to determine the combustion completeness. Fuel moisture consumption thresholds parameters for the 1-hr fuels are updated from (Thonicke et al., 2010) with modifications to the minimum- and mid-moisture thresholds and low-moisture coefficient derived from (Peterson and Ryan, 1986) to remove a drop in combustion completeness at mid-moisture levels (Table 1, Figure S2).

| Table 1. Fuel class characteristics used in the parameter file for this study. Bulk density for dead leaves from (Andrews, 2018) and for live grass from (Snell, 1979), other values from (Thonicke et al., 2010). 1-hr fuel minimum- and mid-moisture thresholds and low-moisture coefficient derived from (Peterson and Ryan, 1986). | | | | | | |
|---|---|---|---|---|---|---|
| **Parameter** | **Twigs (1-hr)** | **Small branches (10-hr)** | **Large branches (100-hr)** | **Trunk (1000-hr)** | **Dead leaves** | **Live grass** |
| Fuel bulk density (fire_FBD, kg m⁻³) | 15.4 | 16.8 | 19.6 | n/a | 4 | 0.95 |
| Fuel surface area to volume ratio (fire_SAV, cm⁻¹) | 13 | 3.58 | 0.98 | 0.2 | 66 | 66 |
| Low-moisture coefficient (fire_low_moisture_coeff, unitless) | 1.12 | 1.09 | 0.98 | 0.8 | 1.15 | 1.15 |
| Low-moisture slope (fire_low_moisture_slope, unitless) | 0.62 | 0.72 | 0.85 | 0.8 | 0.62 | 0.62 |
| Mid-moisture threshold (fire_mid_moisture, m³ m⁻³) | 0.72 | 0.51 | 0.38 | 1 | 0.8 | 0.8 |
| Mid-moisture coefficient (fire_mid_moisture_coeff, unitless) | 2.35 | 1.47 | 1.06 | 0.8 | 3.2 | 3.2 |
| Mid-moisture slope (fire_mid_moisture_slope, unitless) | 2.35 | 1.47 | 1.06 | 0.8 | 3.2 | 3.2 |
| Minimum-moisture threshold (fire_min_moisture, m³ m⁻³) | 0.18 | 0.12 | 0 | 0 | 0.24 | 0.24 |
| Rate of decomposition transfer (max_decomp, yr⁻¹) | 0.52 | 0.383 | 0.383 | 0.19 | 1 | 999 |
| Fraction of woody biomass transferred to CWD pool (frag_cwd_frac) | 0.045 | 0.075 | 0.21 | 0.67 | n/a | n/a |

**2.1.2.3 Rate of Spread**

Once an ignition event occurs, the potential forward rate of spread ($ROS_f$) (m min⁻¹) is calculated as in (Thonicke et al., 2010) per the equations of (Rothermel, 1972):



$$ROS_f = \frac{I_r \, x_i \, (1+ \theta_w)}{BD_{patch} \, \varepsilon \, Q_{ign}} \qquad (7)$$

where $I_r$ is the reaction intensity (kJ m² min⁻¹) and represents the energy release per unit area of the fire front; $x_i$ is the propagation flux ratio, and represents the proportion of $I_r$ that heats fuel particles to ignition; $\theta_w$ is a wind factor; $\varepsilon$ is the effective heating number, and represents the number of particles heated to ignition temperature; $Q_{ign}$ is the heat of pre-ignition (kJ kg⁻¹), which is the amount of heat required to ignite a given mass of fuel, and $BD_{patch}$ is a weighted average of bulk density across the fuel classes in that patch that are available for burning.


### 2.1.2.4 Fire intensity and area burned

The surface fire intensity ($I_{surf}$)(kW m⁻¹) is then calculated as in (Thonicke et al., 2010):

$$I_{surf} = h \, FC_{patch} \, \frac{ROS_f}{60} \qquad (8)$$


where $h$ (kJ kg⁻¹) is the heat content of fuel set to a default value of 18,000 kJ kg⁻¹ and $FC_{patch}$ (kg m⁻²) is the overall fuel consumption from the fire. Fires with a surface intensity below a user defined minimum energy threshold cannot be sustained and are extinguished. The default value for this threshold is 50 kW m⁻¹ per (Peterson and Ryan, 1986; Thonicke et al., 2010). For this study, the minimum energy threshold for sustained burning was set to 25 kWm⁻¹ for sites where the tree canopy cover

is less than a 55% threshold for savanna (Staver et al., 2011) and 75 kWm⁻¹ for areas above this tree cover threshold based on fire intensity measurements for savanna (Govender et al., 2006) and neotropical forests (Brando et al., 2016).

    The total area burned is assumed to be in the shape of an ellipse, with the major axis determined by the forward and backward rates of spread (*ROS$_f$* and *ROS$_b$* respectively).

*ROS$_b$* is a function of *ROS$_f$* and wind speed (*W*):

$$ROS_b = ROS_f \, e^{-0.012W} \qquad (9)$$

The minor axis to major axis ratio, or length to breadth ratio (*l$_b$*) (m), of the ellipse is determined by the wind speed. If *W* is

less than 16.67 m min⁻¹ (i.e., 1 km hr⁻¹) then *l$_b$* =1. Otherwise, *l$_b$* is calculated for forest areas or grass fuel areas using prior values (Forestry Canada Fire Danger Group, 1992; Wotton et al., 2009) based on a forest to grassland threshold per (Staver et al., 2011). Note that there was an typographic error in the *lb* equation for grasses in (Forestry Canada Fire Danger Group, 1992), which was reported and corrected in (Wotton et al., 2009) but nonetheless incorporated into the original SPITFIRE code of (Thonicke et al., 2010); we remove that error and use the (Wotton et al., 2009) equation here. $W_{effect}$ (m min⁻¹) is the

wind adjusted by vegetation fraction with *W* being the site level wind boundary condition.





$$W_{effect} = W \left(tree_{fraction} 0.4 + \left(grass_{fraction} + bare_{fraction}\right)0.6\right) \tag{10}$$


$$lb = \begin{cases} 1.0 + 8.729(1.0 - e^{-0.03 W effect})^{2.155}, & tree_{fraction} > 0.55 \\ 1.1 \, W_{effect}^{0.464}, & tree_{fraction} \leq 0.55 \end{cases} \tag{11}$$

The length of the major axis is calculated for both the front, $d_f$ (m), and back, $d_b$ (m), of the fire ellipse using the associated *ROS*:


$$d_f = ROS_f \, F_{dur} \tag{12}$$

$$d_b = ROS_b \, F_{dur} \tag{13}$$

Fire size, ($F_{size}$) (m$^2$), is calculated using the methods of (Arora and Boer, 2005):


$$F_{size} = \frac{\pi}{4 l_b}(d_f + d_b)^2 \tag{14}$$

The total area burned ($A_{burn,patch}$) (m$^2$ km$^{-2}$) is calculated for fires of size $F_{size}$ (m$^2$) for each of the daily successful

ignitions (km$^{-2}$ day$^{-1}$) ($I_{lightning}$ and $I_{anthro}$) while accounting for the fire danger conditions $FDI$. Ignitions ($I_{lightning}$ and

$I_{anthro}$) are input or calculated for the total gridcell area, and we assume that ignitions are equally distributed per unit area

across each patch; therefore $I_{lightning}$ and $I_{anthro}$ are provided as strikes per km$^{-2}$ of patch area per day. The $A_{burn,patch}$ is

therefore m$^2$ km$^{-2}$ per patch area per day.

$$A_{burn,patch} = F_{size}(I_{lightning} + I_{anthro})FDI \tag{15}$$

**2.1.2.4 Fire damage and mortality**

As in (Thonicke et al., 2010) tree mortality from fire is calculated based on both cambial damage to bark and crown scorch to

the canopy. Damage from crown scorch is calculated in relation to scorch height ($SH$) (m) of a fire:

$$SH = F \, I_{surf}^{0.667} \tag{16}$$


where $F$ is a PFT-specific parameter based on field studies. In this study $F$ is set to 0.1487 for the fire-vulnerable tree and 0.06

for the fire-tolerant tree as in the tropical broadleaved evergreen and tropical broadleaved raingreen tree PFTs respectively

from (Thonicke et al., 2010).





Within FATES, fire effects on plants are evaluated for each cohort that experiences fire. Assuming a cylindrical crown shape, the proportion of crown scorch $CS$ is calculated for each cohort as:

$$CS = \frac{SH - H + CD}{CD} \qquad (17)$$

where $H$ (m) is the height of the tree cohort (m) and $CD$ (m) is the crown depth length calculated using a PFT-specific crown depth fraction ($CD_{frac}$). For this study, the fire-vulnerable tree PFT has a $CD_{frac}$ of 0.33 and the fire-tolerant tree PFT a $CD_{frac}$ of 0.1. The probability of tree mortality from crown scorch ($p_{cs}$) is calculated as:

$$p_{cs} = r(CS^p) \qquad (18)$$


where $r$ is a PFT-specific resistance factor for crown scorch survival and $p$ is a parameter based on defoliation from crown scorch set to a default value of 3.0 (Thonicke et al., 2010). For this study, the resistance factor for crown scorch survival ($r$) is set to 1 for the fire-vulnerable tree PFT and 0.05 for the fire-tolerant tree PFT.

Cambial damage is based on the residence time of the fire ($\tau_f$) and the bark thickness of the cohort. Probability of

mortality from cambial damage ($p_\tau$) is calculated as:

$$p_\tau = \begin{cases} 0.0, & \text{for } \frac{\tau_l}{\tau_c} \leq 0.22 \\ 0.563\,\frac{\tau_l}{\tau_c} - 0.125, & \text{for } \frac{\tau_l}{\tau_c} > 0.22 \\ 1.0, & \text{for } \frac{\tau_l}{\tau_c} \geq 2.0 \end{cases} \qquad (19)$$

where $\tau_c$ is the critical fire residence time (min) based on bark thickness ($BT$) (cm bark per cm DBH).


$$\tau_c = 2.9\, BT^2 \qquad (20)$$

The overall probability of mortality ($p_m$) is calculated as:

$$p_m = p_\tau + p_{cs} - p_\tau p_{cs} \qquad (21)$$


Thus, for each day with a fire, a burned area is calculated for each patch. Fire effects, including consumption of ground fuels, damage to vegetation through cambial damage and crown scorch, are applied to the fraction of each patch that burns, which in turn splits into a newly-disturbed patch with area equal to the area that burned. Fire effects on fuels and vegetation thus only occur on the newly-burned patch. The newly-burned patches resulting from the burned fraction of each patch are given a





time-since-disturbance age of zero and are generally fused together and into other recently-disturbed patches, following the FATES patch fusion logic (Fisher et al., 2015).

### 2.1.3 Model experiments

We defined a series of model experiments reflecting trade-offs associated with fire-tolerance strategies in vegetation traits selected for each plant functional type (PFT), and conducted a test of model sensitivity to the parameter governing fuel drying

(*drying ratio*). We then explored how climate-fuel relationships and vegetation traits mediate ecosystem assembly, as well as the impact of vegetation state on fire behaviour (Table 1). We completed a set of simulations for South America varying the fuel drying ratio, and then compared results to contemporary observations. We then ran a simulation using the intermediate drying ratio parameterization across the tropics.

FATES-SPITFIRE was run as a module within the CLM5 (Lawrence et al., 2019) using air temperature, humidity, wind, air pressure, precipitation and shortwave and longwave radiation produced by the Global Soil Wetness Project (GSWP) for the period 1994-2013, with forcing data at a 6 hourly time step (disaggregated to 30 minute time steps by the native CLM algorithm). The forcing data was part of the third phase of GSWP (GSWP3v1, http://hydro.iis.u-tokyo.ac.jp/ GSWP3/), and is based on the 20th Century Reanalysis version 2 from the NCEP model (Compo et al., 2011). To allow for vegetation to spin-

up, the forcing data was cycled repeatedly for a period of 300 years with the final ten years used for evaluation. All simulations started from a bare ground condition and were conducted under a stable recent historical (2000) $CO_2$ concentration (367 ppm). Anthropogenic land-use was not used in this study, thus these simulations represent a potential vegetation case.

Three PFTs were used in all simulations and allowed to establish and compete on all grid cells: a C4 grass and two tropical

trees, with one tree PFT utilizing a set of fire-tolerant traits and the other a set of fire-vulnerable traits (Table 2). Supplemental seed dispersal from outside the grid cells was disabled in these simulations. Given that coexistence of PFTs is sensitive to the representation of seed rain (Maréchaux and Chave, 2017; Fisher et al., 2010), coexistence is not assured and a PFT may go locally extinct in a given grid cell. The tree PFTs were otherwise parameterized with common traits and allometry from (Koven et al., 2020) with updates to the maximum carboxylation at reference temperature ($V_{cmax}$) (Kattge et al., 2009), leaf longevity

(Kattge et al., 2011), and leaf nitrogen derived from $V_{cmax}$ and specific leaf area (SLA) per the relationship between these quantities derived by (Walker et al., 2014) (model 2 in their Table 3) (Table 2). The tree PFT growth respiration factor (grperc) was adjusted from the CLM5 default of 0.11 to 0.3 for a carbon use efficiency (CUE) with a mean of 50% (Table 2, Figure S3) calculated as the ratio between mean net primary productivity (NPP) and gross primary productivity (GPP). This version of FATES does not use the same maintenance respiration terms as CLM5, and thus gives biased low CUE when the CLM5

value is used. Distinct tree fire strategies represented with trait trade-offs for crown, leaf and bark characteristics were parameterized as in (Thonicke et al., 2010) using their tropical broadleaved evergreen as the fire-vulnerable strategy and their tropical broadleaved raingreen as the fire-tolerant strategy in this study (Table 2). Wood density uses data from (Chave et al.,





2006) (their Table 1) with the lower wood density fire-vulnerable tree represented by the mean value for species from the Amazon forests and the higher wood density fire-tolerant tree represented by the mean value for species from the South

American dry forests (Table 2). Thus, while FATES does not directly impose a penalty on trees for having thick bark or other fire-tolerant traits, we have asserted a trade-off between wood density and fire tolerance, such that in the absence of fire we expect the lower wood density, fire-vulnerable tree to outcompete the higher wood density, fire-tolerant tree.

We estimate grass allometry based on the tiller size and height of *Spartina alterniflora,* which are well studied to parameterize

the allometric relationships in FATES. Grass height allometry is based on observation data from (Daehler et al., 1999) and (Travis and Grace, 2010). The aboveground biomass allometry parameters are estimated based on observed height, tiller diameter and tissue density (Radabaugh, K.R. et al., 2017). The leaf allometry parameters are fitted based on observed aboveground biomass in relationship to height and diameter, assuming that ~50% of above ground biomass is leaf biomass (Gross et al., 1991).  The live fine root biomass to live leaf is set to 1.0 and storage to leaf ratio as 2.25, in view that ~75% of

belowground biomass is rhizome for storage (Schubauer and Hopkinson, 1984). The specific leaf area is estimated from (Giurgevich and Dunn, 1979). The ratio of tiller diameter to crown area is fitted to observed tiller density (Radabaugh, K.R. et al., 2017). The $V_{c,max25}$ is set to 40 umol m$^{-2}$ s$^{-1}$ (Giurgevich and Dunn, 1979).

**Table 2.** Parameter values for the two tree PFTs and C4 grass used in the simulations.

| Parameter | fire-vulnerable tree (Moist_trop_tree) | Fire-tolerant tree (Dry_trop_tree) | C4 Grass |
|---|---|---|---|
| Ratio C store to leaf biomass (storage_cushion, fraction) | 1.2[*] | 1.2[*] | 2.25 |
| Diameter to leaf biomass allometry intercept (allom_d2bl1) | 0.1266844[*] | 0.1266844[*] | 0.000964 |
| Diameter to leaf biomass allometry slope (allom_d2bl2) | 1.281329[*] | 1.281329[*] | 1.9492 |
| Maximum DBH to area factor (allom_d2ca_coefficient_max) | 0.768654[*] | 0.768654[*] | 0.03 |
| Minimum DBH to area factor (allom_d2ca_coefficient_min) | 0.768654[*] | 0.768654[*] | 0.01 |
| Diameter to height allometry intercept (allom_d2h1) | 57.6[*] | 57.6[*] | 1 |
| Diameter to height allometry slope (allom_d2h2) | 0.74[*] | 0.74[*] | 1 |
| Allocation of carbon root per leaf (allom_l2fr, gC gC$^{-1}$) | 0.4863088[*] | 0.4863088[*] | 1 |
| Leaf area per sapwood area intercept (allom_la_per_sa_int, m$^2$ m$^{-2}$) | 0.8[*] | 0.8[*] | 1000 |
| Ratio of SAI per LAI (allom_sai_scaler, m$^2$ m$^{-2}$) | 0.1[*] | 0.1[*] | 0.0012 |
| Branch turnover time (branch_turnover, yr) | 75[*] | 75[*] | 0.3208 |
| Leaf longevity (leaf_long, yr) | 1.4025[#] | 1.4025[#] | 0.3208 |
| Maximum specific leaf area (leaf_slamax, m$^2$ gC$^{-1}$) | 0.03991654[*] | 0.03991654[*] | 0.0135 |
| Top of canopy specific leaf area (leaf_slatop, m$^2$ gC$^{-1}$) | 0.01995827[*] | 0.01995827[*] | 0.0135 |
| Vcmax (leaf_vcmax25top, μmole CO$_2$ m$^{-2}$ s$^{-1}$) | 41[&] | 41[&] | 40 |
| Target N/C concentration of organs (prt_nitr_stoich_p1, gN gC$^{-1}$) | 0.026748659[@] | 0.026748659[@] | 0.16 |
| Growth respiration factor (grperc, unitless) | 0.3 | 0.3 | 0.11 |



| Soil moisture threshold for drought mortality (non-hydraulic version) (Hf_sm_thresh, unitless) | 0.025[*] | 0.025[*] | 1e-06 |
|---|---|---|---|
| C starvation mortality rate (mort_scalar_cstarvation) | 0.02955703[*] | 0.02955703[*] | 0.2 |
| Initial height new plant (recruit_hgt_min, m) | 1.3[*] | 1.3[*] | 0.5 |
| Initial seedling density (recruit_initd, stems m$^{-2}$) | 0.2[*] | 0.2[*] | 20 |
| Fraction C to seeds (seed_alloc, fraction) | 0.04680188[*] | 0.04680188[*] | 0.1 |
| Fraction C to seeds, mature plants (seed_alloc_mature, fraction) | 0[*] | 0[*] | 09 |
| Leaf fire vulnerability (alpha_SH, m kw$^{-1}$ m$^{-1}$) | 0.1487[§] | 0.06[§§] | n/a |
| Bark thickness (bark_scaler, fraction) | 0.0301[§] | 0.1085[§§] | n/a |
| Crown depth (crown_depth, fraction) | 0.33[§] | 0.1[§§] | n/a |
| Crown mortality probability (crown_kill) | 1[§] | 0.05[§§] | n/a |
| Wood density (wood_density, g m$^{-3}$) | 0.6305[¶] | 0.695[¶¶] | n/a |
| References: [*]: Koven et al 2020; [#]: Katgee et al 2011; [&]: Katgee et al 2009; [@]: Calculated based on $V_{cmax}$ and SLA per Walker et al 2014 model 2 their Table 3; [§]: Tropical Broadleaved Evergreen (Thonicke et al 2010); [§§]: Tropical Broadleaved Raingreen (Thonicke et al 2010); [¶]: Mean for species from the Amazon forests (Chave et al 2006); [¶¶]: Mean for species from the South American dry forests (Chave et al 2006) | | | |

A total of five CLM-FATES simulations were completed with four at the 0.5° x 0.5° grid resolution for South America exploring a range of fuel drying ratio parameterizations and one pantropical simulation at the 0.9° x 1.25° grid resolution applying the intermediate fuel drying parameterization. The fuel drying ratio and geometry determine how fuel moisture content responds to fire-relevant weather conditions (Figure S1) and this fuel moisture in turn impacts the effectiveness of combustion (Figure S2) with smaller or drier fuels experiencing more combustion than larger or wetter fuels. To explore the

model sensitivity to this crucial aspect of fire dynamics and allow us to generate potentially variable fire regimes, we modified the parameter for the fuel drying ratio using a value for low fuel drying at 66,000 °C$^{-2}$ (Thonicke et al., 2010), high fuel drying at 13,000 °C$^{-2}$ (Lasslop et al., 2014), and medium fuel drying at 25,000 °C$^{-2}$ (Table 3). In these idealized experiments, we investigated whether fuel drying acts as a significant factor in the biogeography, and explored the connection between fuel drying across the same climate conditions to investigate the span of potential responses across the tropics. In the real world,

these connections will have a more complex and heterogeneous spatial pattern related to variability in local conditions. The simulations were compared against a control simulation without fire disturbance and against contemporary observations.

**Table 3:** Model simulations

| | Fire activity | Fuel Drying Ratio (°C$^{-2}$) | Region | Resolution |
|---|---|---|---|---|
| control | no | n/a | South America | 0.5° x 0.5° |
| Low fuel drying | yes | 66,000 (Thonicke et al 2010) | South America | 0.5° x 0.5° |
| Medium fuel drying | yes | 25,000 | South America | 0.5° x 0.5° |
| High fuel drying | yes | 13,000 (Lasslop et al 2014) | South America | 0.5° x 0.5° |
| Medium fuel drying | yes | 25,000 | Pan-tropical | 0.9° x 1.25° |



**2.1.4 Evaluation data**

We evaluated simulated output using data processed and regridded to 0.5° x 0.5° resolution available as part of the ILAMB project (Collier et al., 2018). Productivity was evaluated using gross primary productivity (GPP) from the GBAF product derived from FluxNet MTE observations (Jung et al., 2010) and leaf area index (LAI) generated from the MODIS satellite observations (De Kauwe et al., 2011). Biomass was evaluated against the carbon stock product of (Saatchi et al., 2011). Simulated burned area was evaluated against the burned area product from the Global Fire Emissions Database (GFED4S,

(Giglio et al., 2013; van der Werf et al., 2017), which includes small fires.

**3 Results**

**3.1 Influence of fuel drying assumptions**

The mean burned area across South America displays a variable spatial pattern among the three FATES simulations that differ in the fuel drying parameterization (Figure 1). The peak burned area region in both the observations and the model extends

from the northeast of Brazil and along the southeastern edge of the Amazon, with high burned areas also to the north of the forest. The three simulations show the high parameter sensitivity of FATES-SPITFIRE, where increased fuel drying leads to simulations with higher burned area than observed. Higher fuel drying parameterizations were associated with an increased spatial extent of burned area, a longer peak fire season with more ignitions, faster forward rate of spread, more intense fires, and higher burned fraction across all months (Table 4), but with the largest increases from June to October (Figure 2). Across

the South American region, the different parameterizations of fuel drying result in different ecosystem structure and function due to the changes to the fire regime, including lower biomass and tree cover with higher fuel drying (Figure 3, Table 4).

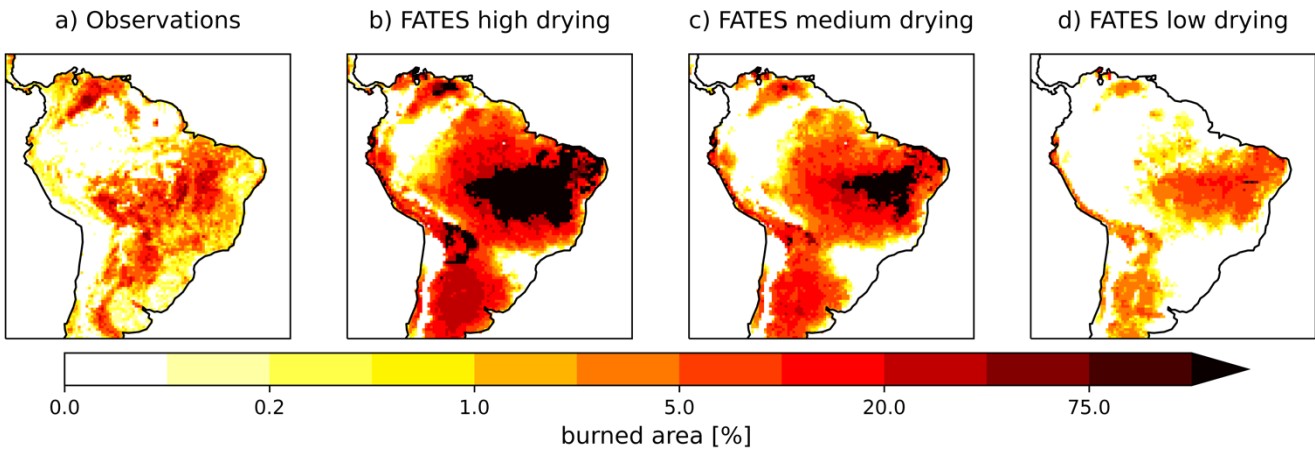

**Figure 1. Mean annual fraction area burned from (a) observations (van der Werf et al 2017) and for CLM-FATES for the final ten years of 300 year simulations with active fire disturbance and a (b) high, (c) medium, or (d) low fuel drying parameterization.**



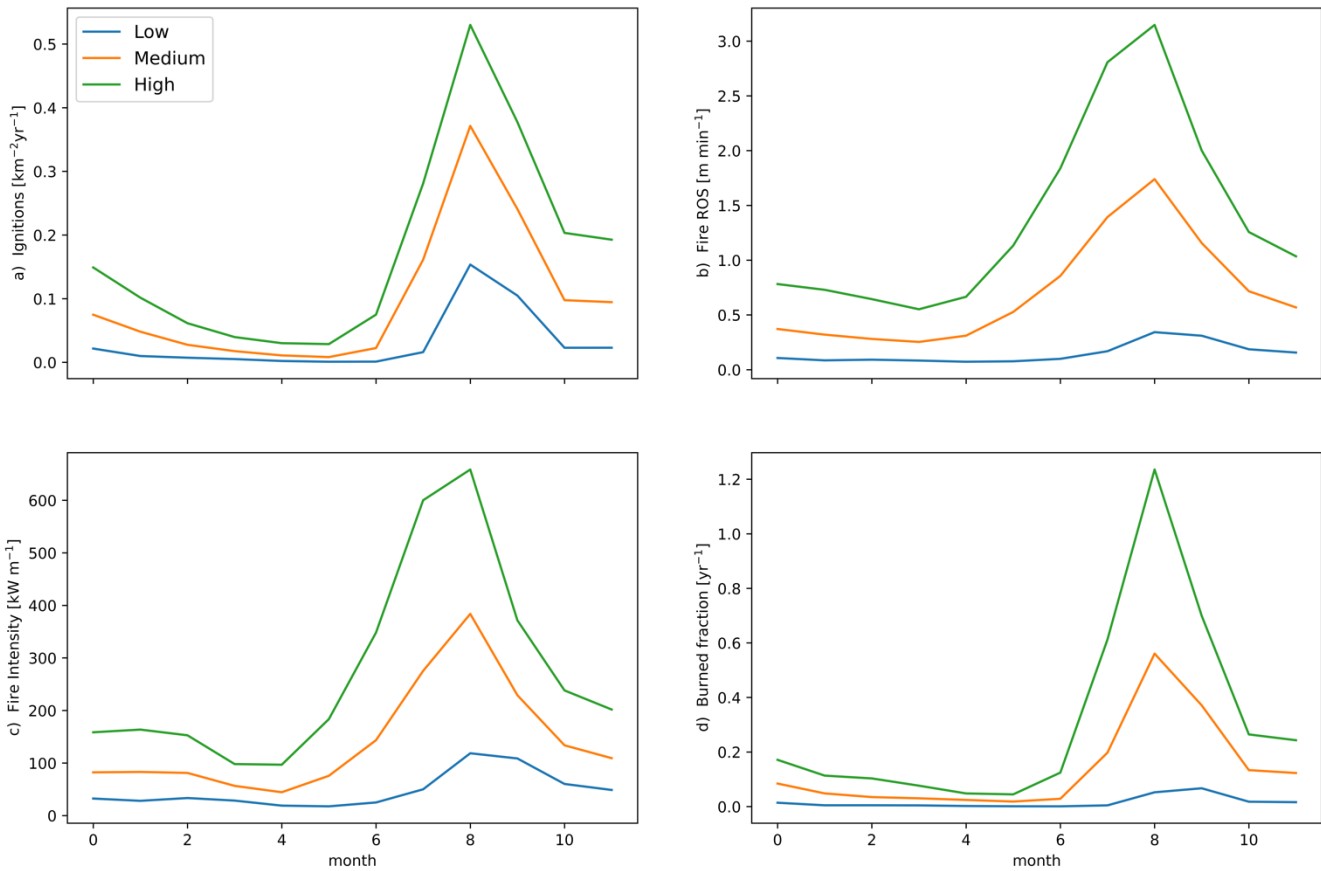

**Figure 2. Mean seasonal change in (a) fire ignitions, (b) rate of spread (ROS), (c) intensity and (d) burned fraction for parameterizations with low (blue), medium (orange) or high (green) fuel drying for the final ten years of 300 year simulations in CLM-FATES across South America.**





**Figure 3. Difference between the high and low fuel drying parameterizations for (a) maximum temperature, (b) minimum temperature, (c) relative humidity, (d) simulated aboveground biomass, (e) tree area, (f) live grass, (g) burned fraction, (h) fire intensity, (i) rate of spread, and (j) ignitions for the final ten years of 300 year simulations in CLM-FATES.**





Though the same atmospheric forcing data was used for all simulations, the resulting vegetation distribution differences under
the high (vs low) fuel drying parameterization led to higher maximum and minimum temperature by as much as 1°C and lower
mean annual relative humidity by up to 4% (Figure 3, S4, S5). These differences were primarily concentrated in regions with
a change in tree cover fraction. Across all simulations, areas which lost biomass were associated with lower relative humidity,
more burning and fire effects (Figure 3, S4, S5). Seasonal declines in precipitation and relative humidity coincide with the

peak fire season, and the highest rates of burning and fire effects occurred in August (Figure 2, S6). The natural seasonal
decline in fuel moisture for dead leaves and live grass coincides with the increase in fire behavior and effects from June to
October (Figure 2, 4), whereas twigs and small branch fuels did not have large seasonal fluctuations in moisture. Across the
region, more intense and larger simulated fires were associated with the presence of live grass fuels, but did not have a clear
relationship with live grass moisture or amount (Figure 5, S7).  Fire intensity decreases as dead fuel moisture increases with

precipitation and relative humidity, but dead fuel amount shows mixed relationships across climate variables, without simple
linear consequences for fire intensity or burned fraction (Figure S8, S9). Mean aboveground biomass decreased with increased
fuel drying, with biomass losses occurring in the drier north-eastern regions of South America (Figure 6, S10, Table 4).

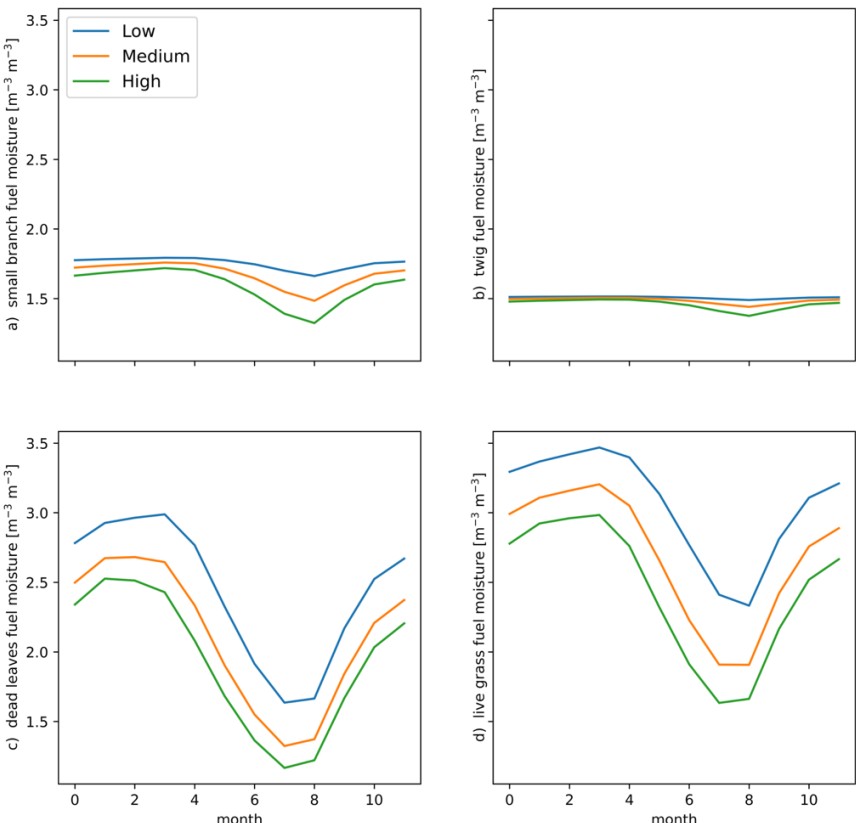

**Figure 4. Mean seasonal change in fuel moisture (m3 m-3) for (a) small branches, (b) twigs, (c) dead leaves, and (d) live grass fuels**
**for parameterizations with a low (blue), medium (orange) or high (green) fuel drying ratio for the final ten years of of 300 year**
**simulations in CLM-FATES.**

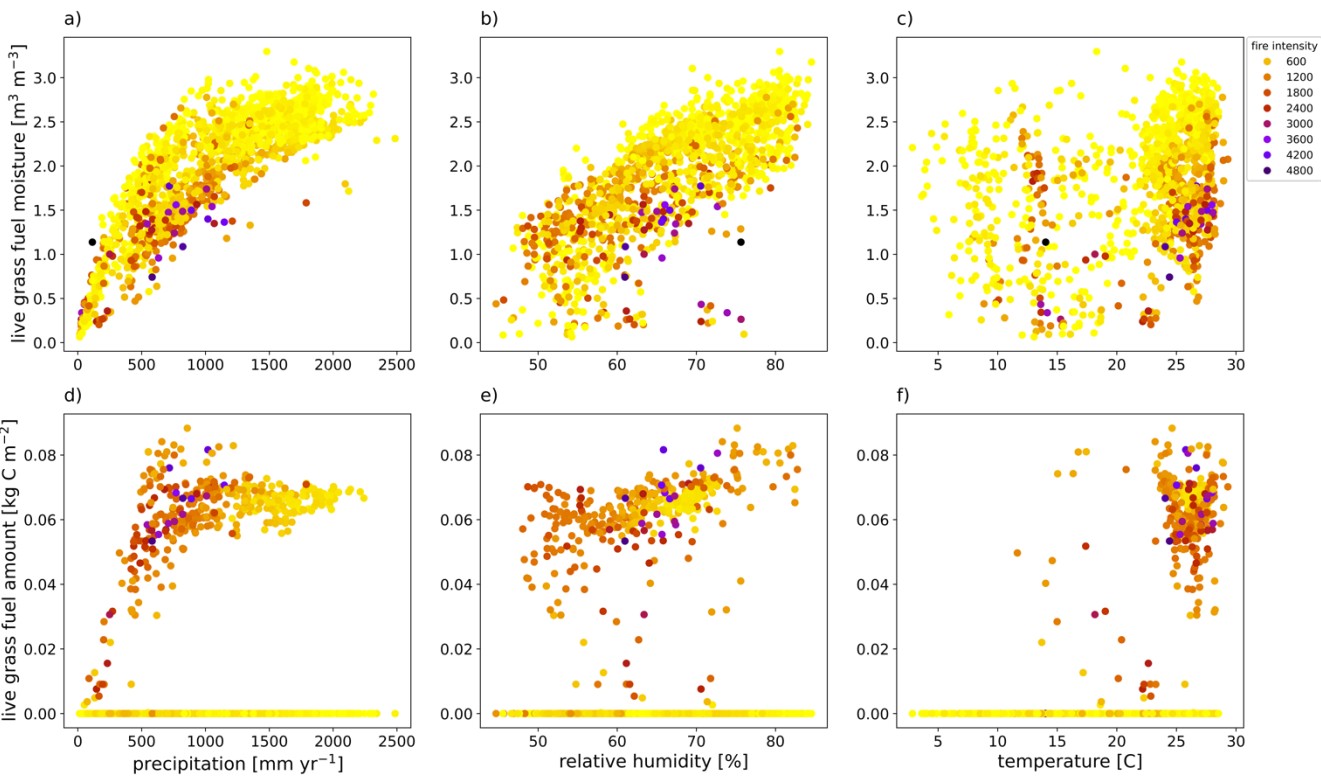

**Figure 5. Association of fire intensity (colors; kW m-1) for live grass fuel moisture (m3 m-3) with (a) precipitation, (b) relative humidity, and (c) temperature, and for live grass fuel amount (kgC m-2) with (d) precipitation, (e) relative humidity, and (f)**
**385  temperature for fire intensities above 100 kW m⁻¹ from the final ten years of a 300 year simulation in CLM-FATES across South America using a medium fuel drying parameterization.**

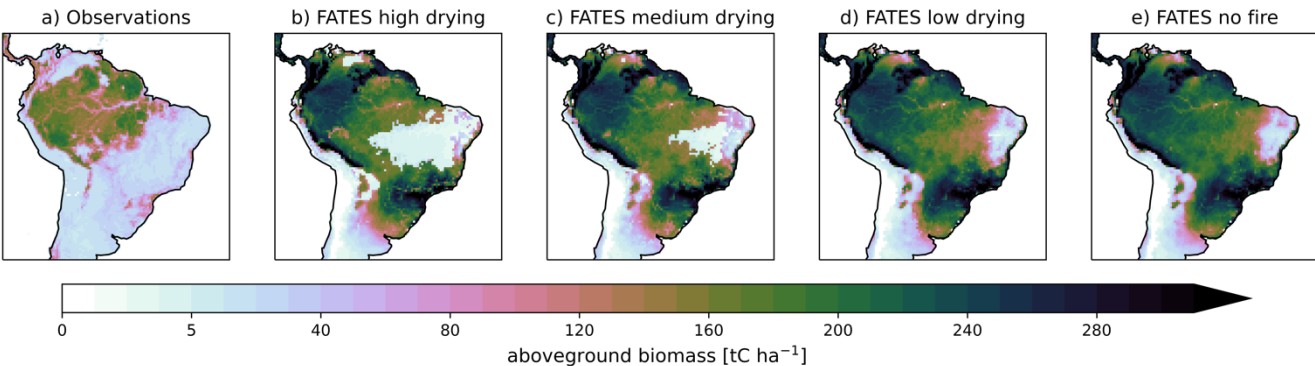

**Figure 6. Aboveground biomass for (a) observations (Saatchi et al 2011), and for CLM-FATES using parameterizations with (b) high, (c) medium or (d) low fuel drying, and (e) without fire disturbance for the final ten years of 300 year simulations.**


Parameterizations with higher fuel drying resulted in the expansion of grass and fire-tolerant tree PFT distributions and their associated biomass, and a lower total mean biomass across the region (Figure 7, Table 4). Comparisons of simulated size-

based fire mortality showed that, for all simulations with fire disturbance, the fire-tolerant trees escaped fire mortality through height and fire resistant traits more effectively than the fire-vulnerable trees, but trees below 20 cm diameter at breast height (DBH) for both PFTs experienced elevated mortality during fire events (Figure 8, S11, S12).

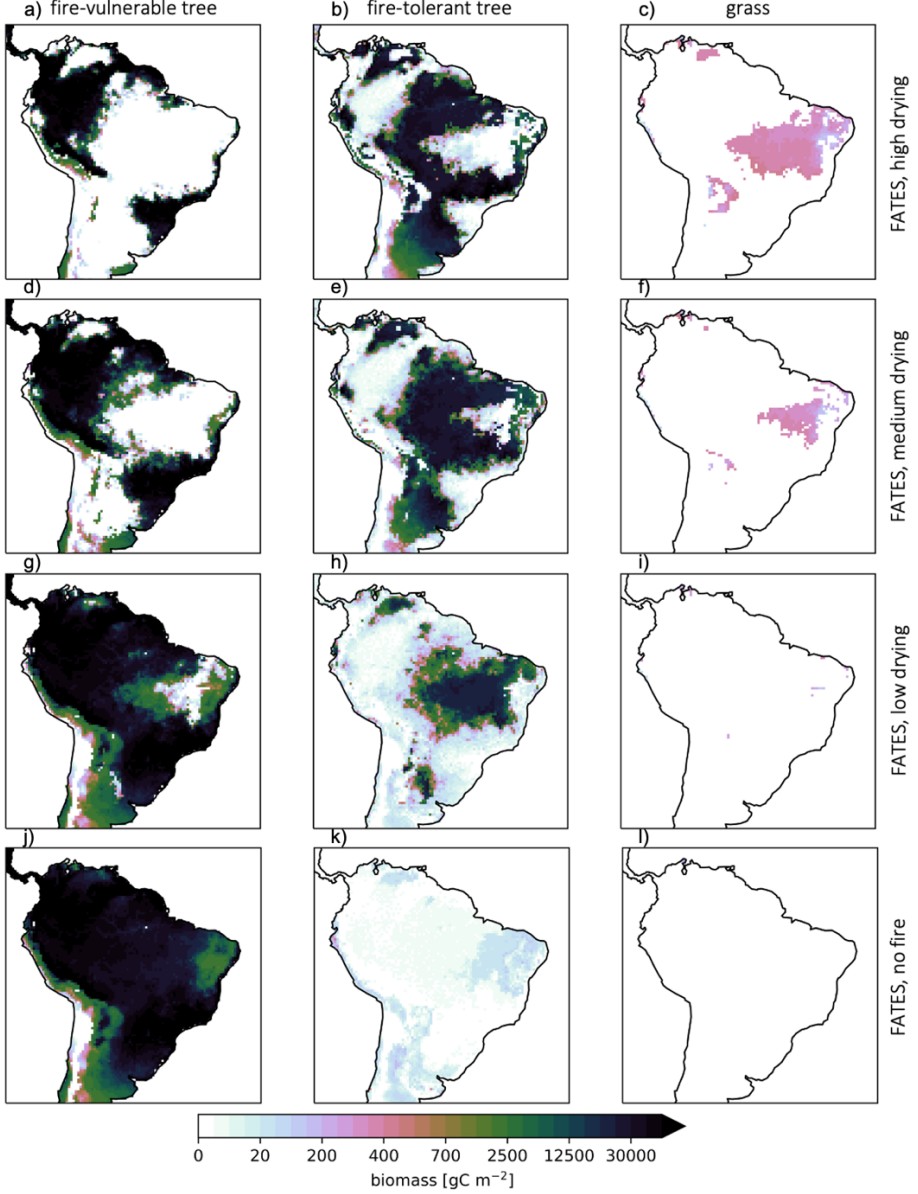

**Figure 7. Mean PFT biomass (gC m-2) for parameterizations with a high (a-c), medium (d-f) or low (g-i) fuel drying and active fire, or no fire (j-l) for the final ten years of a 300 year CLM-FATES simulation.**



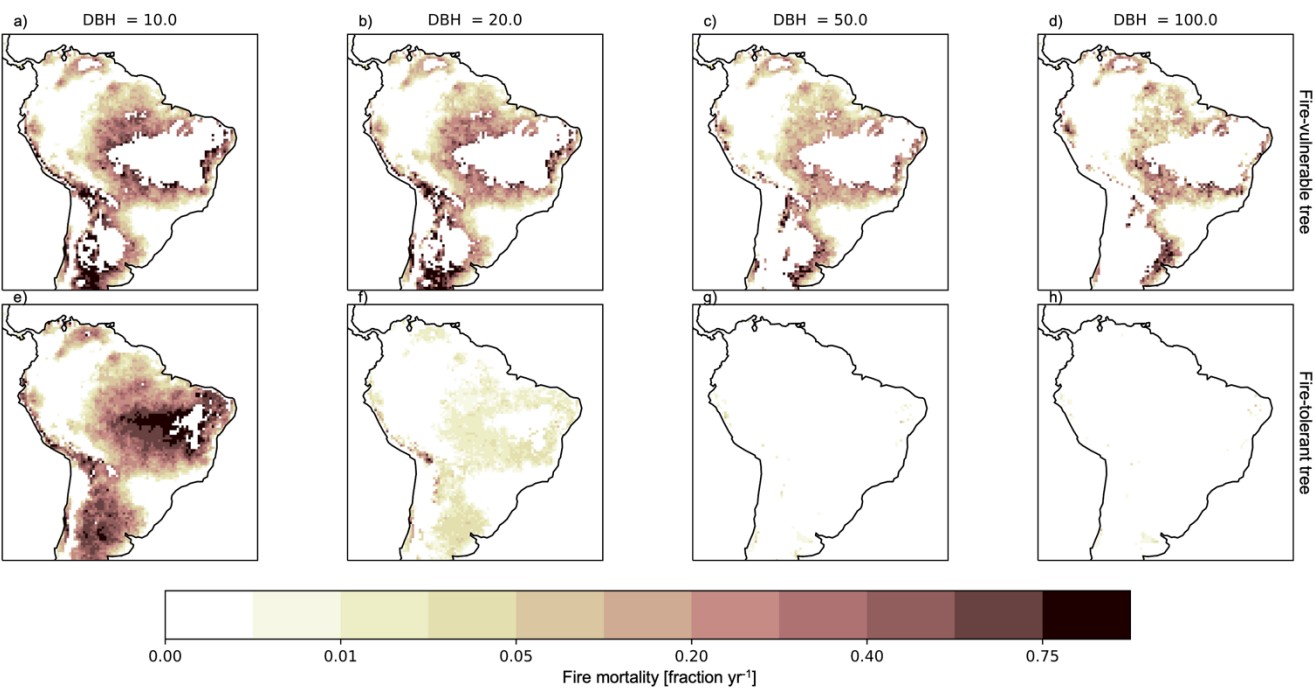

**Figure 8. Mean annual fraction of tree mortality due to fire effects across tree-cohort sizes (diameter at breast height, DBH, of (a, e) 10, (b, f) 20, (c, g) 50 and (d, h) 100 cm) from FATES simulations using a medium fuel drying parameterization for the final ten years of a 300 year simulation. Top row (a-d) is the fire vulnerable tree PFT and bottom row (e-h) is the fire-tolerant tree PFT.**

| Table 4. Mean (maximum, standard deviation) across fuel drying ratio assumptions for South America regional simulations. | | | |
|---|---|---|---|
| **Variable** | **Low fuel drying** | **Medium fuel drying** | **High fuel drying** |
| Aboveground biomass (tC ha$^{-1}$) | 166.37 (368.2, 95.26) | 153.74 (368.2, 96.77) | 136.6 (368.2, 102.7) |
| Leaf Area Index (m$^2$ m$^{-2}$) | 4.92 (7.5,1.67) | 4.36 (7.63, 1.94) | 3.69 (7.48. 2.13) |
| Gross Primary Productivity (gC m$^{-2}$ yr$^{-1}$) | 2307.6 (5750.8, 998.32) | 2299.8 (5924.4, 1036.55) | 2362.6 (6046.18, 1127.2) |
| Burned area (fraction yr$^{-1}$) | 0.0156 (1.120, 0.0391) | 0.1357 (3.188, 0.765) | 0.3069 (8.84, 1.20) |
| Intensity (kW m$^{-1}$) | 49.39 (5536.4, 219.97) | 142.93 (5198, 628.95) | 272.03 (6295.2 912.32) |
| Rate of spread (m min$^{-2}$) | 0.1556 (24.11, 0.799) | 0.7101 (31.55, 3.25) | 1.377 (24.71, 4.14) |
| Ignitions (km$^{-2}$ yr$^{-1}$) | 0.0304 (0.4085, 0.0543) | 0.09733 (0.8643, 0.359) | 0.1722 (1.141, 0.470) |
| Temperature max (degree C) | 30.17 (35.89, 4.96) | 30.18 (36.12,5.66) | 30.20 (36.17, 5.68) |
| Temperature min (degree C) | 17.98 (26.32, 6.16) | 18.10 (26.32,6.64) | 18.23 (26.35, 6.65) |
| Relative humidity (%) | 75.88 (93.31, 11.67) | 75.43 (93.30, 16.69) | 74.94 (93.29, 16.75) |
| Precipitation total (mm) | 1615 (9029, 126.4) | 1615 (9029, 126.4) | 1615 (9029, 126.4) |





### 3.2 Comparisons against observations


Active fire disturbance across the South American region reduced the biomass density (Figure 6, 9). When comparing our simulations of a potential forest state to contemporary observations we find that all simulations, including the no fire simulation, had a high bias compared to contemporary observations for biomass (Figure 9), especially in the less disturbed areas of the Amazon and, as expected given the lack of land use in the simulations, in the highly anthropogenically-disturbed

Atlantic coastal forest region (Figure 6). Without fire disturbance, the fire-vulnerable tree becomes dominant, driving the grasses to extinction throughout much of the domain, and the fire-tolerant tree to near extinction (Figure 7). Simulated vegetation productivity (GPP), showed a high bias across grassland-dominated regions and a low bias for forested regions when compared to contemporary GPP data products (Figure S13, S14). Mean GPP (gC m$^{-2}$ yr$^{-1}$) and leaf area index (LAI) across South America were high for all fuel drying parameterizations (Table 4), compared to the mean GPP of 1981.6 and the

LAI of 2.68 for observations (Figure S13). The observed seasonality of fires was captured by the medium fuel drying scenario simulation with agreement on the timing of peak fire season (June to October; Figure 2). The simulated burned area across the South American region for the medium and high fuel drying parameterizations had areas of repeat annual burns that were not in the observations and extended into the eastern Amazon, beyond that of observations (Figure 1). Within the forested areas, fires had mean fire intensity values less than 300 kW m$^{-1}$ (Figure S15), which is consistent with the fire intensities observed

in these ecosystems (Brando et al., 2016).

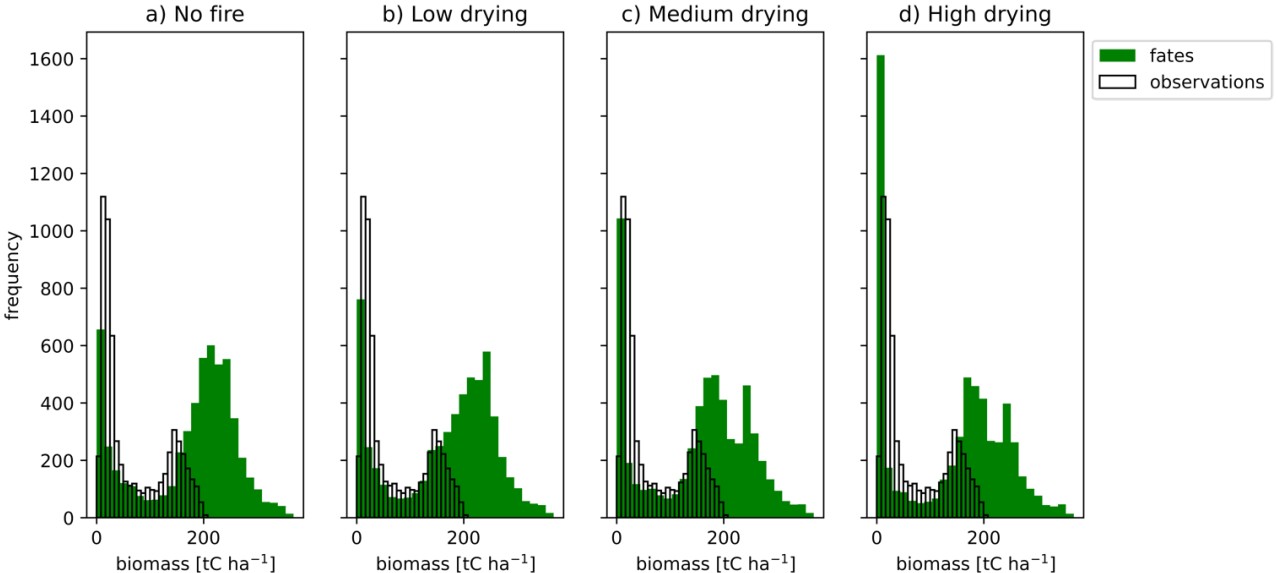

**Figure 9. Mean aboveground biomass (tC ha$^{-1}$) across South America from observations (Saatchi et al 2011) (clear) and for CLM-FATES (green) for the final ten years of 300 year simulations using parameterizations (a) without fire disturbance, and with (b) low, (c) medium, and (d) high fuel drying.**





### 3.3 Pan-tropical application


Application of the medium fuel drying parameterization across the tropics for a 1x1 degree simulation also showed high biases in simulated biomass for areas of naturally occurring high biomass accumulation across wet areas of Africa and Indonesia (Figure 10, 11) with observations 60% lower than the simulated values. Simulated mean annual burned fraction was high, with areas of repeat burns that extended beyond observed burned areas (Figure 12). In the simulation, mean annual rainfall (MAR)

(mm yr$^{-1}$) above 2500 mm is associated with closed forest canopies and nearly continuous tree cover, fire intensity is generally below 150 kW m$^{-1}$, and there is low frequency and extent of burning (Figure 13). Across the tropics, simulated mean GPP, LAI, aboveground biomass, and burned fraction were biased high compared to observations (Table 5, Figure S14). Pantropical simulated burned fractions were associated with grass areas for the highest simulated mean annual fire intensities, generally above 400 kW m$^{-1}$ (Figure 12, 13).

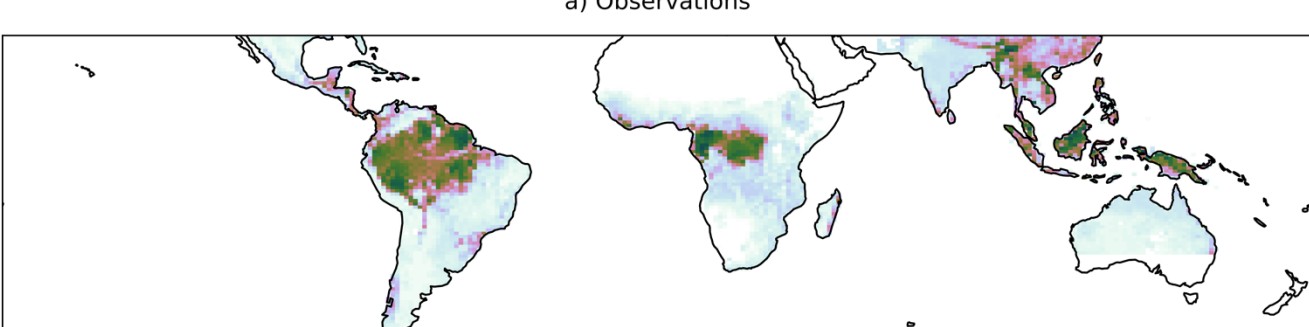

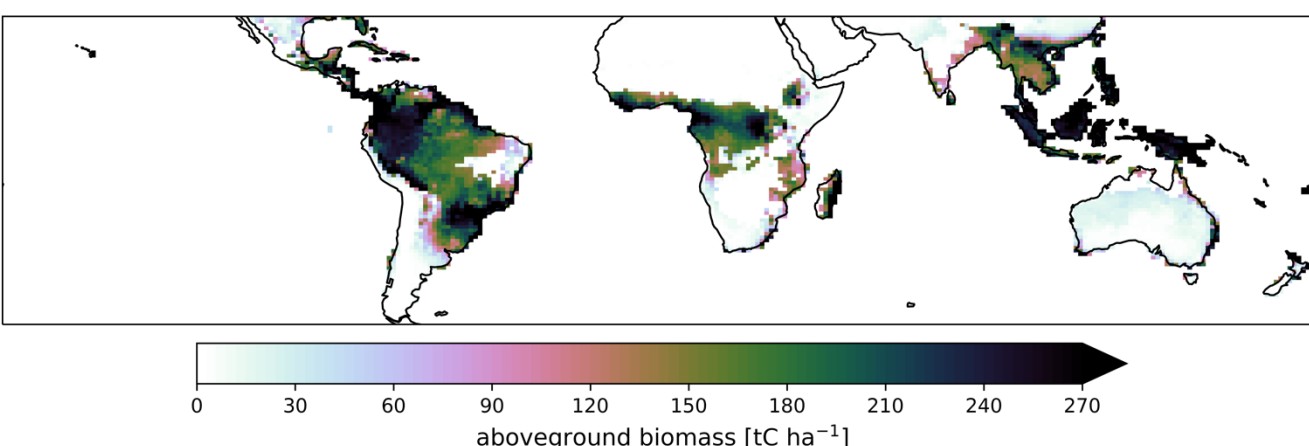


**Figure 10. Mean aboveground biomass from (a) observations (Saatchi et al 2011) and (b) CLM-FATES from the final ten years of a 275 year simulation with active fire disturbance and medium fuel drying parameterization.**



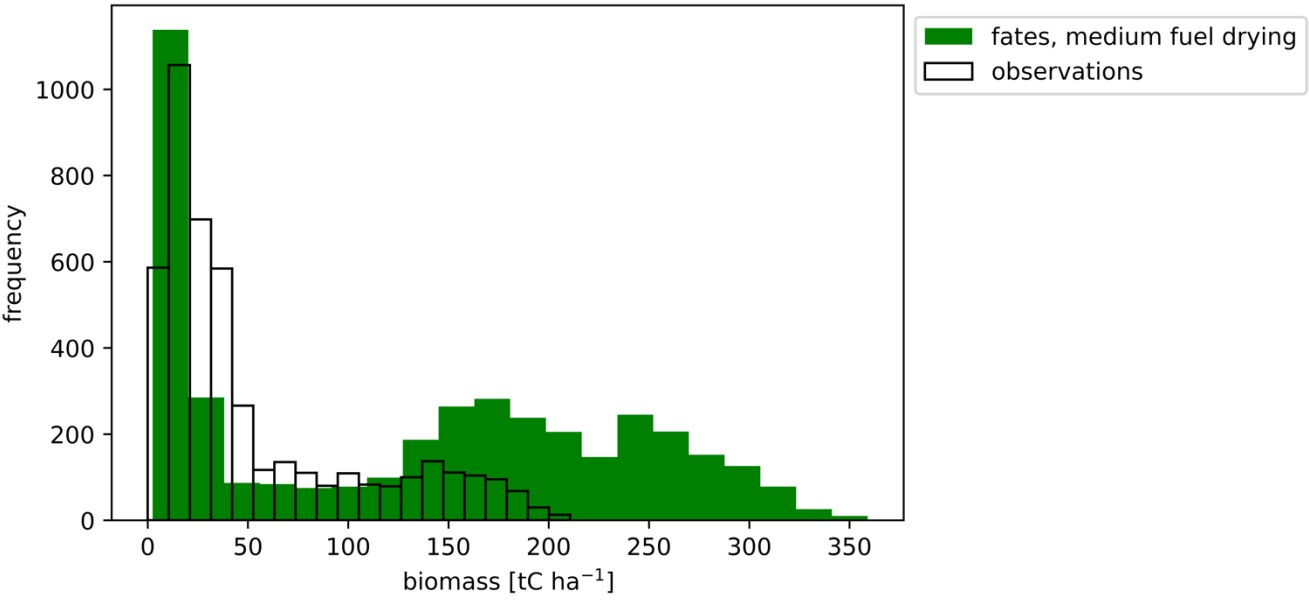

**Figure 11. Mean aboveground biomass (tC ha-1) for the pan-tropics from observations (Saatchi et al 2011) (clear) and CLM-FATES (green) for the final ten years of a 275 year simulation with parameterizations for a medium fuel drying ratio.**




**Figure 12. Mean burn area (% yr-1) from (a) observations (GFED41s) and CLM-FATES simulation for (b) burned area, (c) fire intensity, and (d) C4 grass biomass from the final ten years of a 275 year simulation with active fire disturbance and medium fuel drying parameterization.**





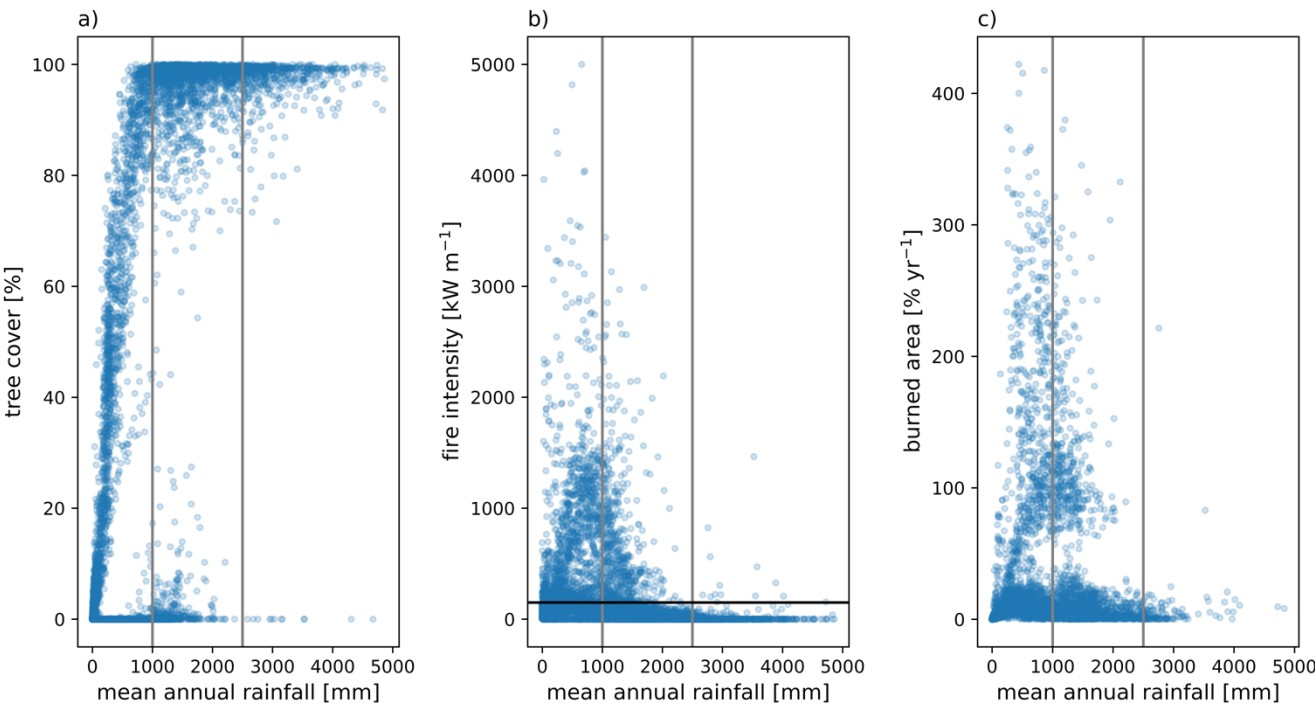

**Figure 13.** Simulated mean (a) tree cover, (b) fire intensity, and (c) burned area as a function of mean annual rainfall (MAR) (mm yr-1) from the final ten years of a 275 year CLM-FATES simulation across the tropics with active fire disturbance and a medium fuel drying parameterization. Grey vertical lines indicate 1000 and 2500 MAR. Horizontal black line in (b) indicates a fire intensity of 150 kW m-1.

**Table 5.** Mean (maximum, standard deviation) across the tropics for a CLM-FATES simulation with a medium fuel drying parameterization and from observations.

| Variable | Medium fuel drying | Observations |
|---|---|---|
| Aboveground biomass (tC ha-1) | 87.48 (359.46, 104.12) | 52.04 (210.68, 51.81) |
| Leaf Area Index (m$^2$ m$^{-2}$) | 2.73 (7.46, 2.38) | 1.306 (5.88, 1.399) |
| Gross Primary Productivity (gC m$^{-2}$ yr$^{-1}$) | 1670.3 (6158.2, 1384.5) | 1026.8 (3177.36, 904.9) |
| Burned area (fraction yr-1) | 0.278 (37.13, 1.26) | 0.0118 (11.983, 0.152) |
| Intensity (kW m-1) | 275.83 (59663.5, 889.36) | |
| Rate of spread (m min-2) | 1.90 (351.14, 5.13) | |

## 4 Discussion

Globally, fire disturbance and associated fire behavior and effects are important contributors to shifting ecosystem structure and function (Bowman et al., 2020; McLauchlan et al., 2020). We demonstrate here that size-structure and fire-tolerance strategy together determine the susceptibility of trees to fire mortality and the resulting biogeography and accumulation of biomass. Further, the FATES-projected biomass and distribution of simulated fire-tolerant and fire vulnerable trees and grasses





were strongly influenced by fuel drying and associated fire behavior, highlighting the importance of fuel state interacting with fire-tolerance traits to structure savanna and forest biomes.

Small tree cohorts of both types suffered high mortality in fire-prone areas, but fire-tolerant trees were more consistently resilient across fuel parameterizations, including in simulations with drier fuels that resulted in increased fire frequency. The

variation in functional strategies was fundamental to capturing shifts in vegetation type and overall biomass accumulation across a gradient of fire disturbance. Fire-tolerant trees and grasses are more competitive under increased fire conditions, and conversely, impeded via resource-related competition dynamics under fire-free conditions.

## 4.1 Tropical biogeography and associated fire behavior

### 4.1.1 Vegetation traits and size-structure as drivers of fire behavior and effects

Across a wide range of fire intensity and frequency, fire acts as a selective pressure on tree survival and ultimate success. Though all simulated small trees experienced high mortality across the fuel drying parameterizations (Figure 8, S11, S12), it was only through trait differences that tree biogeography was determined. The trade-off between wood density and fire tolerance (Table 2) provided a competitive advantage to the fire-vulnerable tree in areas where fire was absent. Simulated biogeography for the medium fuel drying parameterization, where the fire-vulnerable, large-canopied, thin-barked tree was

dominant across the Amazon region (Figure 7), reflects the spatial distribution of thin versus thick bark trees documented by (Pellegrini et al., 2017). These results are in agreement with studies of bark variation and its association with fire disturbance in the tropics (Staver et al., 2020; Pellegrini et al., 2017; Hoffmann et al., 2009; Brando et al., 2016; Uhl and Kauffman, 1990) in suggesting that fire-driven losses will be higher in fire-vulnerable forests. Thus, representation of the spatiotemporal dynamics of competition between PFTs with different fire tolerance strategies is critically important for prediction of future

fire severity impact on vegetation biomass accumulation and composition.

Differences in fire mortality across tree sizes is well documented (Hoffmann et al., 2009; Uhl and Kauffman, 1990; Brando et al., 2016), and our results are consistent with previous studies. Fire effects on trees are generally dependent on the trees' size-structure, bark thickness and canopy characteristics. For studies at the edge of frequently fire-disturbed areas in South America,

on the rare occasions that fire enters forests, smaller trees are killed, but larger trees survive (Hoffmann et al., 2009; Higgins et al., 2000; Hoffmann and Solbrig, 2003) based on the larger trees accumulation of thicker bark and a taller canopy height that escapes flame damage. Among established fire-tolerant trees, once a tree has surpassed the height of the flame-zone, mortality is low in fire disturbed areas (Higgins et al., 2000; Hoffmann et al., 2009). Our results capture the decrease in tree mortality with an increase in size (Figure 8, S11, S12). In our model, the simulated fire-tolerant trees were able to maintain a

distribution of biomass and basal area across sizes (Figure S16, S17) in the fire disturbed area, despite experiencing high mortality for plants smaller than 20 cm DBH (Figure 8, S11, S12). The fire-vulnerable tree, in contrast, became extinct in these


fire prone areas (Figure S16, S17). Field studies on fire disturbance impacts on vegetation structure and function are limited, with only two main field studies in the dry Amazon forests of Tanguro, Brazil (Brando et al., 2016, 2012) and the Cerrado forests of the IBGE Reserve (Hoffmann et al., 2009). Increased survivability past 20 cm DBH in simulations is consistent with
field measurements by (Balch et al., 2015; Brando et al., 2016) for experimental burns in the dry forests of Tanguro.

Fire mortality in FATES results from the combination of bark and canopy effects that vary as trees grow larger, with canopy damage from crown scorch calculated as a function of fire intensity and PFT-specific fire tolerance (EQ 16). A shift to drier fuels leads to an increase in simulated mean fire intensity to 143 kW m$^{-1}$ for the medium fuel drying parameterization from 49
kW m$^{-1}$ for the low fuel drying parameterization (Figure S15, Table 4). This shift to drier fuels extends fire associated tree mortality into the Amazon (Figure 8, S12) implying that the increase in fire intensity under the medium fuel drying parameterization surpasses the mortality threshold for these fire-vulnerable trees. The simulated mean fire intensity of 143 kW m$^{-1}$ for the medium fuel drying parameterization is close to the fire intensity threshold value of 149 kW m$^{-1}$ derived by (De Faria et al., 2021) using data from (Staver et al., 2020) demonstrating that bark thickness continues to increase and protect
against mortality until reaching this fire intensity threshold. The pattern of increased tree mortality for areas with simulated fire intensities beyond the fire intensity threshold derived by (De Faria et al., 2021) suggests broad agreement with the functional relationship from (Staver et al., 2020). The Amazon has high diversity among trees classified as fire-vulnerable or fire-tolerant and within that diversity tree bark thickness varies in space and time due to its connection with demography, and these results only capture two broad categories of trees with variable strategies. Simulation results from the low fuel drying
parameterization, with a mean fire intensity of 49 kW m$^{-1}$, still demonstrated mortality across all sizes for the fire-vulnerable tree (Figure S12), with mortality occurring within the Amazon region, but at a much lower rate compared to simulations using increased fuel drying parameterizations (Figure 8, S11). This allows the simulated fire-vulnerable tree to extend its dominance across much of South America through competitive advantage. Across the tropics, fire-tolerant trees vary from one savanna region to another, and are not a subset of forest tree species (Bond et al., 2003; Hoffmann et al., 2003). Within Australia, the
Myrtaceae family, which includes eucalypts and is characterized as excellent resprouters even after high-intensity fire, dominates fire-dependent forests across the continent (Crisp et al., 2011; Burrows, 2002) and may actually promote fire with elevated fuels and flammable leaf litter (Lehmann et al., 2011). These continental differences among trees are not accounted for in this current study, and suggest that for forested regions outside of South America further parameterization and additional PFTs may be needed to capture the interaction between climate, fire and vegetation.


Further, this version of FATES does not include the capacity of trees to resprout following fires, which is a key feature of persistence for trees in savanna regions (Gignoux et al., 1997; Hoffmann et al., 2009; Govender et al., 2006; Higgins et al., 2000). In reality, fires often cause loss of the whole aboveground stem, but not mortality for some individual fire-tolerant trees. This "topkill" of the individual tree stem is then followed by resprouting, which can accelerate recovery after fires (Hoffmann
et al., 2009; Higgins et al., 2000; Van Wilgen et al., 2004). The actual rate of mortality can therefore be low in established fire-





tolerant stands where small trees are able to persist even with repeated topkill by fires (Higgins et al., 2000; Hoffmann, 2000). Additionally, simulated trees in FATES can experience canopy and/or bark cambial damage from fire without mortality, but damaged trees in FATES do not experience a long-term loss of function and quickly return to a pre-fire state with pre-disturbance allometry. Across the tropics, crown damage is an important predictor of mortality (Reis et al., 2022; Arellano et al., 2021), and after light limitation, crown damage was the most important mortality risk indicator (Zuleta et al., 2022). Future work will link fire-related damage to the FATES mechanistic crown damage module (Needham et al., 2022) that imposes limitations on tree regrowth following a damage event as the trees attempt to recover.

Within the closed forests of the Amazon, grasses are not typically present as the dense canopy shades them out (Hoffmann et al., 2003; Brando et al., 2020; Cochrane et al., 1999), but along the drier savanna and Cerrado regions there is the possibility for trees and grasses to co-exist (Hoffmann et al., 2003; Higgins et al., 2000). Across the tropics, mean annual rainfall (MAR) acts to moderate tree cover which limits fire behavior at levels beyond 50% tree cover (Staver et al., 2011; Pueyo et al., 2010). Intermediate MAR between 1000 and 2500 mm can be associated with both forest, with tree cover greater than 55%, or savanna, with less than 55% tree cover, though there is variability across continents (Staver et al., 2011; Hirota et al., 2011). Though our simulations capture this range of variability in tree cover for low to intermediate MAR (Figure 13), we do not capture a large degree of coexistence between trees and grasses. Grass-tree coexistence is thought to be driven by limited opportunities for trees to escape drought and flame zones before entering larger size-classes (Higgins et al., 2000), so it is at these transitional stages where grasses might invade into the forests. Thus, changes in demography and regeneration are essential in exploring tree-grass coexistence, where slower tree growth rates or consistent disturbance allows grasses to invade and expand, but faster tree growth rates favor forest expansion (Higgins et al., 2000). FATES captures only very limited areas of tree-grass coexistence at the edges between the fire-tolerant tree and C4 grass area, as the simulated forest canopy quickly closes and shades out the simulated grasses (Figure 7). Fire behavior varies depending on vegetation type, number and distribution of patches and surface fuels, with all of these interacting to generate overall fire behavior and effects. For this study, in areas with less than 55% tree cover we use a grass-specific fire spread equation (EQ 11), where for the same wind speed low tree cover and grass areas will have a longer length to breadth ratio for the resulting burned ellipse than higher tree cover areas (Wotton et al., 2009). This threshold for the shift in the shape of the resulting burned area warrants further investigation as it may be reinforcing feedbacks to promote grasses and prevent tree-grass coexistence.

The simulated fire-tolerant tree is resilient under the higher fuel drying parameterizations, maintaining and expanding its area of dominance into the range of the fire-vulnerable tree, while losing range in drier regions to the C4 grass with associated fire increases. The loss of simulated biomass of 17.14 tC ha$^{-1}$ across the whole domain with a transition from fire-vulnerable trees to fire-tolerant trees and grasses highlights the vulnerability of regional carbon stores with drier fuels (Table 4). The potential for biomass loss associated with increased fire disturbance is in agreement with previous studies (De Faria et al., 2021; Burton et al., 2022; Bond et al., 2005). Notably, areas that are degraded by disturbance have demonstrated colonisation by grasses that



then facilitate increased fire frequency and expanded grass invasion (Balch et al., 2015; Veldman and Putz, 2011; Silvério et al., 2013; Hoffmann et al., 2012; Veldman et al., 2009). Conversion from forest to grasses has been shown to dramatically increase fine fuel loads compared to forest litter (Silvério et al., 2013; Hoffmann et al., 2012), thereby increasing the potential for more intense fires. Our results demonstrate this increase of fire intensity with the presence of grasses (Figure 5, 12). Historically, fire was a major factor in determining the current distribution of grasses (Bond and Midgley, 2012b; Staver et al.,

2011; Hirota et al., 2011; Lehmann et al., 2011; Sankaran et al., 2005; Bucini and Hanan, 2007; Sankaran et al., 2008). Slow tree recovery after fire has also been suggested as a key factor in the spread of grasses (Bond, 2008; Hoffmann et al., 2009), as fires maintain grasses in areas suitable for forests through frequent burning that favors vegetation with underground storage (Bond and Midgley, 2012b; Ratnam et al., 2011; Hoffmann, 2000). At forest-savanna margins trees are able to recruit and grow when they escape the influence of grass fires through local variations in seasonal fire intensity (Higgins et al., 2000;

Balch et al., 2015; Govender et al., 2006).

### 4.1.2 Fire intensity and fuel dynamics

The spatial patterns in simulated fire intensity and burned fraction were defined through interactions between the climate and emergent vegetation and climate, resulting in, broadly, two groups: low intensity fires below 150 kW m$^{-1}$ in regions without grass presence and higher intensity fires (> 500 kW m$^{-1}$) in regions with grass presence (Figure 12, 13). The simulations

demonstrated a positive grass-fire feedback where regions with MAR below 2500 mm have frequent and high intensity large fires that promote grass dominance. Simulated fire intensities for these areas with low to intermediate MAR (less than 2500 mm) (Figure 12, 13) are consistent with those measured in savannas in Kruger National Park in Africa (up to 17905 kW m$^{-1}$) (Govender et al., 2006), the Northern Territory of Australia (500 -18000 kW m$^{-1}$) (Williams, et al., 2003), the Campos grasslands of Brazil (36 to 319 kW m$^{-1}$) (Fidelis et al., 2010), and the Cerrado of South America (2842 to 16394 kW m$^{-1}$)

(Kauffman et al., 1994). Across the tropics with sufficient moisture and in the absence of grazing, grass production during the wet season becomes available as fuel during the dry season capable of supporting frequent fires (Higgins et al., 2000; Bond and Midgley, 2012b; Govender et al., 2006). The seasonal shifts in fuel availability and moisture for fine fuels of live grass and dead leaves demonstrated in this study (Figure 4) were associated with higher fire intensities and burned fraction (Figure 5) and are consistent with previous work (Higgins et al., 2000; Balch et al., 2015; Hoffmann et al., 2012; Govender et al.,

2006).  In contrast, regions with high MAR above 2500 mm generally have low simulated fire intensities, with values generally below 150 kW m$^{-1}$, and more than 80% tree cover, characteristics that are consistent with observed understory fires of the Amazon (Figure 13).

Historically, within the Amazon, fire had previously been limited to deforested or agricultural areas (Alencar et al., 2011), as

the closed forest canopy creates a moist understory microclimate environment that limits the potential for fire (Brando et al., 2020; Hoffmann et al., 2009). Understory low-intensity fires are documented across portions of the Amazon (Morton et al., 2013; Aragão et al., 2018), but the range and variation in fire intensity of these understory fires is not extensively documented





(Staver et al., 2020; Cochrane et al., 1999). This study simulated fire intensities across South America with mean values of 49 kW m$^{-1}$, 143 kW m$^{-1}$ and 272 kW m$^{-1}$ for the low, medium and high fuel drying parameterizations, respectively (Table 4).

Simulated fire intensity varied regionally as well as seasonally, but across the Amazon was consistently lower in all fuel drying parameterizations (Figure 2, Figure S15). This places the simulation that used the low fuel drying parameterization within the upper range of intensity values derived for the Amazon by (Staver et al., 2020) using data from (Cochrane et al., 1999) with an upper limit of around 55 kW m$^{-1}$ and below the value of 75 kW m$^{-1}$ reported by (Brando et al., 2016) for Tanguro which is located on the dry edge of the Amazon. These differences between the Amazon and Tanguro suggest that fuel characteristics

for the closed-canopy forests of the Amazon are not the same as the open and sparse canopy forests in drier regions of South America, such as Tanguro. Characterization of fuels, both their presence as live and dead fine fuels, their decomposition, geometry and moisture, are key uncertainties for fire models (Hanan et al., 2022). In the simulation, the fuel drying parameterization is the same across South America, and fire behavior and intensity respond to climate, vegetation and fuel variability. The effects of moist understory microclimate on fuel characteristics as is documented for closed canopy forests are

not captured in this version of FATES, but future versions that include moist understory conditions, such as through the use of a multi-layer canopy (Bonan et al., 2021), may increase fuel moisture and thus lower fire intensity mechanistically with fuel moisture responding to local microclimate conditions rather than a global drying parameterization. The low simulated levels of fire occurrence, burned area, and fire intensity, with energy generally below 150 kW m$^{-1}$ associated with high MAR above 2500 mm for the tropical simulation (Figure 13) demonstrates behavior consistent with that of understory forest fires in the

Amazon where grasses are excluded and fine surface fuel amounts are limited. Furthermore, low intensity understory forest fires, such as those observed in the Amazon, are not representative across the tropics, which is characterised by a diversity of pyromes, or regions with similar fire characteristics, such as regions with high intensity large fires like those found in Australian (Archibald et al., 2013). This version of FATES does not include the potential for a surface fire to become a crown fire, whereby a surface fire ignites canopy fuels creating a more intense fire, but future work will include the potential for

crown fire behavior. Though these results have frequent burning across the tropics (Figure 12), they do not fully capture the potential for high fire intensities across the diversity of forested areas (Archibald et al., 2013).

### 4.2 Modeling fire behavior and effects at the Earth system scale

At the Earth system scale there are an increasing number of models which capture fire occurrence and impacts (Hantson et al., 2016), but they vary in the process complexity and aspects of fire that are included (Hantson et al., 2020). The Fire Model

Intercomparison Project (FireMIP) is an international initiative aimed at comparing and evaluating existing models against benchmark datasets at the global scale (Hantson et al., 2016; Rabin et al., 2017; Forkel et al., 2019). Many of these models, like FATES, use simplified processes, such as aggregated area burned rather than individual fires and fire spread, due to challenges in representing the complexity of how fire behavior changes from the flame scale to fire event and the coupling and interactions between those scales (Hantson et al., 2020). Similar to the reductions in tree area when including fire seen in this

study, a multi-model global assessment of fire-induced tree cover change demonstrated a consistent reduction with the most





significant losses in savanna regions with low tree cover and high burned area compared to simulations without fire disturbance (Lasslop et al., 2020). Current fire-enabled dynamic vegetation models demonstrate the largest amount of uncertainty with the representation of how the anthropogenic impact on fires is characterized (Teckentrup et al., 2019). The inclusion of anthropogenic impacts and their improved characterization within global fire models is an important uncertainty and

opportunity for fire-enabled models like FATES (Jones and Tingley, 2022; Teckentrup et al., 2019; Venevsky et al., 2019; Forkel et al., 2019; Chuvieco et al., 2021). (Lasslop et al., 2020) demonstrated that parameterization of fire impacts, rather than the extent of burned area in fire-vegetation models, explains more of the differences between them. Capturing the regional variability in the timing and type of human impacts on fires, and the resulting impact on ecosystems remains a challenge for current fire-enabled models that must be addressed (Teckentrup et al., 2019; Venevsky et al., 2019). Expanding and

encouraging further development in fire-enabled dynamic vegetation models is an important step towards improving our ability to represent future fire behavior and effects in a fully coupled ESM. With the current representation of anthropogenic impacts, (Burton et al., 2022) used the dynamic vegetation model JULES-ES to demonstrate that across South America under various future scenarios with increases in temperature and $CO_2$ the increases in burned area and reductions in biomass and tree area imply that there is the potential for enhanced drying across the region. The representation of vegetation-fire-climate feedbacks

are crucial to exploring these types of land-atmosphere interactions.

Our study focuses specifically on the trait-tradeoff between fire-tolerant and fire-vulnerable trees and their competition with C4 grass in changing conditions of fuel dryness. The FATES-SPITFIRE fire module includes impacts on size-structure, fuel and fire characteristics, as well as fire behavior in a dynamic framework; all of which are critical components to capturing fire

behavior and effects in this system (Balch et al., 2015; Brando et al., 2012; Cochrane et al., 1999). The use of generalized PFT parameters is not meant to capture detailed site-level responses, but rather potential biogeography across the region. The dynamic vegetation-fire feedback as displayed through shifts in biogeophysical and biogeochemical properties and fire behavior in response to vegetation shifts (Figure 3, S4, S5) highlight the utility of this framework in exploring feedbacks and interactions. Simulated size-structured mortality and fire intensity across the Amazon and South America were representative

of observations (Brando et al., 2016; Balch et al., 2015; Hoffmann et al., 2009), suggesting that FATES is capturing the mechanism of size-structured mortality for the region (Figure 8). Though the results demonstrated a high bias for biomass accumulation in areas of low disturbance, in fire-disturbed areas the importance of these competitive trade-offs were evident through the variable dominance of PFTs across different fuel drying parameterizations (Figure 7) in relation to increased fire intensity and burned area (Figure S15). Our results showed increased fire behavior and effects with a transition to grasses that

supports an increase in flammable fine fuels and fire intensity, and is consistent with field measurements (Hoffmann et al., 2012; Balch et al., 2015; Brando et al., 2012; Higgins et al., 2000; Govender et al., 2006; Williams, et al., 2003; Kauffman et al., 1994). Our work is also consistent with that of a process-based model of forest growth and fire effects in demonstrating that the drier parts of the Amazon are vulnerable to grass conversion in response to changing disturbance drivers (De Faria et al., 2021). Additional modelling work has considered the balance of trees and grasses in tropical forest-savanna-grassland





transition areas, and all agree that fire is an essential factor in simulating the dominance of grasses in fire-prone areas (Scheiter and Higgins, 2009; Bond et al., 2003; Bond and Midgley, 2012b; Baudena et al., 2015; Blanco et al., 2014). Though our study does not examine the dynamic response to climate change, the variable fuel drying parameterizations provide a proxy for fuel response to altered climate (Figure S1) and suggest that drier fuels support increased dominance of grassland. Conversion of forest to grassland may increase the risk of fire susceptibility in regions globally (Bowman et al., 2020). Further, the association

of grasses with higher fire intensities and the high rate of size-related mortality for fire intensities above 150 kW m$^{-1}$ suggests that a return to a forest state after grass conversion may be challenging for fire-vulnerable trees of the Amazon.

The incorporation of size-structure and its interaction with a process-based fire behaviour and effects module, as in this study, adds a level of complexity that allows for improved exploration of the impacts of fire and vegetation structure on ecosystem

resilience and functioning. Under the same climate forcing, drier fuels promoted increases in fire behavior that allowed grasses to establish and support regular fire occurrence in areas suitable for tree establishment. The results support a positive grass-fire feedback, and are in agreement with modeling studies from an array of models of variable complexity (Baudena et al., 2015; Blanco et al., 2014; De Faria et al., 2021; Bond et al., 2005). Previous studies suggest that C4 grasses established and expanded under conditions of fire and low $CO_2$ conditions (Scheiter et al., 2012; Higgins and Scheiter, 2012), and that

increasing $CO_2$ potentially favors trees over C4 grasses (Bond et al., 2003; Bond and Midgley, 2012a). Elevated $CO_2$ increases carbon assimilation and plant water use efficiency (Swann et al., 2016), and thus could favor C3 plants, but drought may offset this by maintaining or increasing flammability (Bowman et al., 2020). FATES is well positioned to explore the interaction between $CO_2$, drought and flammability through the process-based representation of fire, interaction between above- and below-ground processes impacting soil moisture, and simulation of leaf-level responses to altered $CO_2$. Further, though there

are general structural similarities, the interactions and feedbacks of fire on savanna and grassland systems varies across continents through variation in vegetation traits (Bond et al., 2003; Hoffmann et al., 2003) and below-ground site conditions, therefore vegetation response to altered climate and fire disturbance should not be assumed to be consistent across regions (Scheiter et al., 2013; Buis et al., 2009; Bond et al., 2005; Lehmann et al., 2011).

Fire has a clear role in determining the biogeography of forests, savannas and grasslands across the tropics. Shifts in vegetation type and structure across the Amazon have implications for the global coupled climate system with impacts on water cycling and climate regulation through altered albedo, increased drying associated with forest degradation,  decreased resilience and carbon sequestration capacity of Amazon forests (Artaxo et al., 2022b; Hubau et al., 2020; Artaxo et al., 2022a; Lawrence et al., 2022). This work advances our ability to capture dynamic ecosystem assembly and the potential for shifts in vegetation

state and structure in response to climate-fire-vegetation feedbacks. The Amazon forest ecosystem is coupled across scales via feedbacks between vegetation and climate – a significant conversion of forest affects regional climate, which then feeds back on forest ecosystems, potentially driving further degradation. Future modeling work that captures changes in climatic conditions at forest edges and within stand microclimate would help to incorporate the influence of forest degradation which





is increasing globally (Brando et al., 2019; Baccini et al., 2017; Silva Junior et al., 2020). Capturing this in a modelling context
would allow more detailed exploration of the interaction between land clearing activities, such as logging or agricultural
conversion, and forest degradation, and could quantify potential future impacts of degradation on carbon cycling and forest
flammability under scenarios of deforestation and altered climate. Representing fire in the context of interactions between the
social environment, the physical environment and policy sphere is an essential advance for the current generation of fire-
enabled land surface models to better inform and support global communities (Shuman et al., 2022).

**5 Conclusion**

Because FATES explicitly tracks size-based competition and mortality and the feedback between vegetation and fire, it can
be used to explore the response of the system to fire under variable conditions. The results presented here demonstrated a long-
term response of the system to consistent fire disturbance under stable climate and $CO_2$ conditions without anthropogenic
influences. The mechanisms demonstrated in this study provide a foundation for exploring the impacts of each of these factors
on vegetation biogeography and fire behavior and effects. Results suggest that drier fuels promote a positive grass-fire
feedback, increased fire behavior and characteristics, and an overall loss of biomass, as fire-tolerant vegetation with lower
biomass accumulation rates have a competitive advantage under increased disturbance. Increased fire intensity and area burned
are associated with areas that have less than 2500 mm of annual rainfall, whereas higher rainfall regions have consistently
higher tree cover and low intensity surface fires characteristic of understory fires observed in the Amazon. Though the
simulations capture appropriate size-structured tree mortality due to low intensity fires, these results highlight the need for the
incorporation of crown fire behavior to capture the potential for high intensity fires observed in regions such as Australia. This
study further confirms that vegetation traits associated with fire-tolerance adaptations, size level interactions, and vegetation-
fire feedbacks are important in capturing the response of ecosystems to fire disturbance.

**Code and data availability**

The code is available at https://github.com/jkshuman/fates/tree/fates_api15.0.0_crown_scorch_damage . The parameter file,
analysis scripts and output files are archived at https://doi.org/10.15486/ngt/1992487 . GSWP3 data used to force the model is
available at https://hydro.iis.u-tokyo.ac.jp/GSWP3/

**Supplement**

The supplement related to this article is included.



**Author contributions**

JS, RF, CK and RK developed the code. JS set up the model, performed simulations, and prepared the figures. CX prepared the C4 grass parameterization. JS performed the analysis with inputs from RF, CK and LK. JS wrote the paper with contributions from all co-authors.

**Competing interests**

The authors declare that they have no conflict of interest.

**Acknowledgements**

JKS was supported by the National Center for Atmospheric Research, a major facility sponsored by the National Science Foundation (NSF) under Cooperative Agreement no. 1852977. JKS, RF, CK, RK, LK and CX were supported as part of the Next-Generation Ecosystem Experiments – Tropics, funded by the US Department of Energy, Office of Science, Office of Biological and Environmental Research. RF also acknowledges funding by the European Union's Horizon 2020 (H2020) research and innovation program under Grant Agreement No. 101003536 (ESM2025 – Earth System Models for the Future) and 821003 (4C, Climate-Carbon Interactions in the Coming Century). We would like to acknowledge high-performance computing support from Cheyenne (doi:10.5065/D6RX99HX) provided by NCAR's Computational and Information Systems Laboratory, sponsored by the National Science Foundation.

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
