# Peer review of "Dynamic ecosystem assembly and escaping the "fire-trap" in the tropics: Insights from FATES 15.0.0"

_Geoscientific Model Development, 2023_

## Referee Comment (RC2)

This paper documents the integration of SPITFIRE into FATE within CLM model. It then explores how fire-related traits can influence climate-vegetation-fire interactions using three hypothetical scenarios of different fuel drying parameters mainly over South America. The results are encouraging and show well the importance of representing fire-related ecological traits and integrating size-related fire mortality in land surface models. The paper is generally well written although with some minor errors (see my detailed technical comments). I have some general comments followed by more technical ones listed below:

General comments:

(1) It should be stated clearly that the different drying ratios are used as hypothetical scenarios, simply because only one of them should represent, or be the closest to, the reality. For this reason, I suggest the authors always take caution when they interpret their results by comparing with observations because the model has not yet been fully parameterized and as a result, any disagreements with observations are not a surprise. Rather, the different scenarios are used to explore how fire activity can influence the distribution of forests (and grass) with different fire vulnerabilities.

(2) Table 2 is a bit overwhelming for the readers to know the key differences between the two forest PFTs with different fire vulnerabilities. A short paragraph summarizing these differences with brief explanations on why they represent different fire vulnerabilities would help a lot.

(3) For the key message of the paper, I think using the case of South America would suffice. I don't see a lot additional value by including the simulation for the whole tropics but this makes the whole manuscript more complex. I hence suggest simply removing the results for tropics. This will not reduce the value of the paper. In addition, the model is not subject to a full parameterization, which makes correct simulation over the whole tropics unlikely. Lines 512-514 also refer to the continental differences that are not accounted for in the current parameterization of the model. One may not expect a model to represent the gradient across tropics by using two tree PFTs with different vulnerability. As has also been pointed out by the authors in lines 530-545, the model cannot reproduce the co-existence of tree and grass over the tropics. Compared to Fig. 1 in Staver et al. (2011), Fig. 13a failed to reproduce the large spread in tree cover for the medium rainfall of 1000-2500 mm. Also, the increase in tree cover with rainfall is much steeper in Fig. 13a. The last factor is that tropical forests are subject to anthropogenic fire, deforestation and forest degradation, which are not accounted for in this study.

(4) For model validation: do you use observation corresponding to the simulation period? This is not mentioned in the manuscript but I may have neglected it.

(5) A short paragraph in the Methods describing how forest dynamics related to disturbance are represented in FATE would be helpful for the readers to better understand the work. For lines 268-269: I don't understand whether tree mortality was simulated as partial mortality or treated as stand-replacing mortality. In the latter case, I guess, say, a 10% mortality is simulated, you just take this 10% area and start it with a new patch?

(6) The discussion is in general a little excessive given the hypothetical nature of the study. For example, lines 530-545 focus on comparing tree cover distributions with the observation but from the above we know that many reasons contribute to the disagreement; lines of 622-634 on the anthropogenic effects of fire seem not relevant with this study.

Technical comments:

Line 69–70: it would be nice the authors could expand the discussions here and give a brief description of the status quo on how the effects of tree size and bark thickness on fire-caused mortality are represented, or not represented, in current fire models embedded in land surface models. I understand this needs a bit of work but it can provide a nice overview.

Line 125: regarding the Nesterov Index. I checked that in Thonicke et al. 2010 daily maximum temperature, rather than daily temperature, was used. But in Venevsky et al. (2002), indeed daily temperature was used. So changes are needed here: (1) at least avoid citing both when using daily temperature but rather explain clear the usage in both citations and explain clearly which one was used here. (2) could you explore a bit the influence by following Venevsky rather than Thonicke et al.? This might seem minor but it would be nice to do this.

Line 113: starting from the fact that the data was used in Li et al. is not a good justification. Maybe just say that you used this dataset and Li et al. also used it?

Line 116: "under favorable conditions for burning": a crucial detail. What do you mean by 'favourable' here? How is it implemented in the model? Do you use a constant of 10% or this percentage changes with something (e.g., FDI)?

Line 133: "Weighted averages across fuel classes": what has been used a weight? There are two options, one can either use surface-to-volume ratio or fuel weight.

Line 133-135: does FATE allow tree growing over grasses? The description here seems that CWD, dead litter and live grass form an integrated vertical profile. This is also relevant to lines 529-531. Fig. 7 shows biomass for different PFTs rather than their grid coverage and hence it is unclear whether the ground coverage of different PFTs also show a similar pattern.

Line 139: does 'fine' fuels mean leaf litter? Does FATE represent branches of different diameters so that it is straightforward to classify dead branches into different wood fuel classes or you need some allocation/partitioning scheme?

Line 145: define what is fuel moisture, because some studies use water/(water+dry fuel), some use water/(dry fuel)

Line 11&12 in Supplement: Eq xx, please check

Line 13 in Supp: what do you mean by 'fuel moisture consumption'?

Eq. 3 and Eq. 4: explain what 'fc' stands for to increase readability.

Line 166: the authors assume the reader know the meaning of 'drying raito' by default but I don't understand

it.

Eq. 4: define SAV_fc

Line 156: $\propto$fc, $\propto$ is a strange symbol. Is it the same sign as infinitive? Could you use something easier to understand, read and remember? What do you mean by 'user-defined'? does it mean that for every model application, e.g., in different regions, this parameter needs to be parameterized ? Or it just means that it is parmaterizable?

Lines 145-165: Overall the readability around the fuel moisture simulation part is poor. Could you improve it? More detailed descriptions will allow others to reproduce your research more easily.

Table: citation of Table 1 is poor. Some names in the first column are strange and I don't know where they are used and what they are for, e.g., Rows of 3-9.

Line 189-191: do you check how many fires were actually suppressed/extinguished due to this threshold?

Line 199: $l_b$ has no unit? $L_b$ should be defined as 'major to minor axis ratio' (consistent with Eq. 14), the text gives the reverse.

Line 202-204: could you please detail about this error? Is it about Eq. (10) or the second half of Eq. (11)? Your second half of Eq. (11) is also different from Thonicke et al. (Eq. 13 in their articles).

Eq. (15): a critical detail here: combing back to my comment on Line 116, is $l_{lightning}$ here exclusively scaled by 10%?

Eq. (19): I found an error in the middle sub-equation when implementing spitfire in the recent ORCHIDEE trunk version. 0.563*0.22= 0.12386, which is smaller than 0.125. So there is chance that you might get negative value from this equation which is not plausible. I suggest replacing 0.125 by 0.12386 to avoid this. I found this because it prompts mass balance error in ORCHIDEE due to a negative value.

Line 284-285: was fire module switched on from the very beginning of the simulation? I don't know if there is a risk that trees are too small at the very beginning and they get repeatedly and easily killed by fire. Do you have this issue?

Fig. 2: are these results (panels a, b, c) averaged only for days with fire occurrence? How should I understand panel d? for the green line, if I accumulate the values across all months, does it mean that all grid cells have been almost twice (with accumulative value is about 2 judging by eyes)?

Fig.3: what are these temperatures? Land surface? Surface air? Fire flame?

Table 3: I suggest changing 'low fuel drying' to 'low fuel aridity', 'medium fuel drying' to 'medium fuel aridity'. Because there is the parameter of 'fuel drying ratio', using 'drying' in both of them makes easy confusing.

Table 4: Adding maximum value makes the table a lot more brain-consuming to read. Is it really necessary? Are the data mean values across all grid cells + years?

Fig. 4: Better to use the name of fuel class (1hr, 10hr, …) rather than 'small branch', 'twig' because the latter gives an impression that we indeed have these being represented in the model but actually we are partitioning the biomass into different fuel classes.

Line 373: still, from the top panels of Fig. 5, there is a tendency of higher intensity with lower live grass fuel moisture?

Line 302-304: What's the role of wood density in the model or how this relates to the simulation of fire or vegetation processes? This question is also relevant for lines 467-468.

Line 497: 'mortality threshold' in this line implies something quite precise but I don't think there is any threshold in the model to determine whether a tree was killed completely or not. The mortality is simulated as a continuous number (fraction) indicating mortality rate? No?

---

## Author Comment (AC2)

**Dynamic ecosystem assembly and escaping the "fire-trap" in the tropics: Insights from FATES_15.0.0**

Jacquelyn K. Shuman, Rosie A. Fisher, Charles Koven, Ryan Knox, Lara Kueppers and Chonggang Xu

*Correspondence to*: Jacquelyn K. Shuman ([Jacquelyn.k.shuman@nasa.gov](mailto:Jacquelyn.k.shuman@nasa.gov))

**Response to Review**

We appreciate the review and have updated the manuscript as directed. Those changes are detailed here in blue font below each comment and within the manuscript.

On behalf of the authors,

Jacquelyn Shuman

This paper documents the integration of SPITFIRE into FATE within CLM model. It then explores how fire-related traits can influence climate-vegetation-fire interactions using three hypothetical scenarios of different fuel drying parameters mainly over South America. The results are encouraging and show well the importance of representing fire-related ecological traits and integrating size-related fire mortality in land surface models. The paper is generally well written although with some minor errors (see my detailed technical comments). I have some general comments followed by more technical ones listed below:

General comments:

1. (1) It should be stated clearly that the different drying ratios are used as hypothetical scenarios, simply because

   only one of them should represent, or be the closest to, the reality. For this reason, I suggest the authors always take caution when they interpret their results by comparing with observations because the model has not yet been fully parameterized and as a result, any disagreements with observations are not a surprise. Rather, the different scenarios are used to explore how fire activity can influence the distribution of forests (and grass) with different fire vulnerabilities.

   Text has been added to the introduction at lines 81-82, the methods at line 412, and in the discussion (line 674-675, 697, 870) to more clearly identify this as exploration of fire activity and vegetation-fire feedbacks using hypothetical fuel drying scenarios.

2. (2) Table 2 is a bit overwhelming for the readers to know the key differences between the two forest PFTs with different fire vulnerabilities. A short paragraph summarizing these differences with brief explanations on why they represent different fire vulnerabilities would help a lot.

Text has been added to the methods to detail the trait differences and their impact more clearly for the vulnerability to fire (line 385-388)

3. (3) For the key message of the paper, I think using the case of South America would suffice. I don't see a lot additional value by including the simulation for the whole tropics but this makes the whole manuscript more complex. I hence suggest simply removing the results for tropics. This will not reduce the value of the paper. In addition, the model is not subject to a full parameterization, which makes correct simulation over the whole tropics unlikely. Lines 512-514 also refer to the continental differences that are not accounted for in the current parameterization of the model. One may not expect a model to represent the gradient across tropics by using two tree PFTs with different vulnerability. As has also been pointed out by the authors in lines 530-545, the model cannot reproduce the co-existence of tree and grass over the tropics. Compared to Fig. 1 in Staver et al. (2011), Fig. 13a failed to reproduce the large spread in tree cover for the medium rainfall of 1000-2500 mm. Also, the increase in tree cover with rainfall is much steeper in Fig. 13a. The last factor is that tropical forests are subject to anthropogenic fire, deforestation and forest degradation, which are not accounted for in this study.

We appreciate the reviewer's suggestion to simplify the paper but feel that the inclusion of the full tropical simulation is important to highlight the application potential of FATES as a fire-enabled land surface model, while acknowledging the areas of known improvement. We agree that application beyond the tropics will require additional parameterization and testing, but feel it is within the bounds of this work to apply beyond South America across the tropics. Most of the land surface models applied across the tropics also use a simplified set of species with different vulnerability. Of the fire-enabled land surface models evaluated by FireMIP, there are typically two types of tropical trees represented: tropical evergreen and tropical deciduous, sometimes called tropical broadleaved evergreen and tropical broadleaved raingreen. Seven of the eleven models used two tropical tree PFTs, and the LPJ-GUESS-SPITFIRE model used three tropical tree PFTs (see Rabin et al supplement Tables S17-S28). FATES use of two tropical trees and a C4 grass is within scope of these fire-enabled land surface model applications. The FATES results across South America show reasonable performance for capturing biomass accumulation and broad fire behavior and characteristics for a situation without anthropogenic influence. Global application requires simplification and that we only add necessary complexity needed to answer our scientific questions. Given that the PFT parameterization came from South America, we expect that there is potential lack of agreement beyond South America and include further detail to document that because not all savanna regions or species share these same characteristics as South American species there is reason to look for disparity. These results do demonstrate agreement with field observed fire intensity range among savanna regions across the tropics for areas of low and intermediate rainfall, suggesting that climate limitation on vegetation and subsequent fuel loads may be an important factor. However, though these results suggest reasonable agreement for overall biomass distribution, fire intensity, and frequent burning in these savanna regions, we do not expect agreement for the forest regions of Australia given that Australia has a high intensity active crown-fire regime. The model LPJ-GUESS-SIMFIRE-BLAZE includes additional PFTs for savanna regions distinguishing between Australia and the remaining savannas specifically (Rabin et al 2017). We have included

the detailed information on the variety of fire regimes and tree species adaptations to these fire regimes across the tropical continents because the vegetation tolerance and potential adaptation to fire is an important part of capturing fire-vegetation-climate feedbacks. We highlight Australia specifically because it is an area with high intensity active crown fires in its forests, and active crown fire is missing among nearly all land surface models, and in active development for FATES. Without the inclusion of more diverse tree characteristics and realistic active crown fire dynamics, the ability of a dynamic vegetation model to capture the emergent vegetation of Australia's trees is difficult.

The challenges of capturing tree-grass coexistence, within a mixed demography model with patch level disturbance is related to the canopy closure highlighted by be reviewer and displayed in Figure 13 of the manuscript. We agree that FATES closes the canopy quickly and does not capture the spread of canopy cover as shown in Staver et al (2011) Figure 1. We acknowledge in the manuscript multiple mechanisms for exploring tree-grass coexistence and expect that with progress towards higher tree-grass coexistence FATES will better capture more variability in canopy closure. EQ 12 which is used to characterize the ellipse shape for "grass dominated" fires vs "tree" fires may be acting to promote grasses and prevent coexistence. Alternately, the quick closure of the canopy may be shading out the grasses in areas suitable for coexistence. This manuscript demonstrates that FATES can capture the emergent patterns of biomass as determined by trees and grasses across the tropics. A future study focused primarily on grass-tree coexistence would help to understand the mechanism behind capturing intermediate canopy closure and tree-grass coexistence. We agree with the reviewer that FATES demonstrates canopy closure that is too rapid and are exploring the mechanisms for allowing more diversity in canopy closure. This represents an area of improvement for FATES.

We further agree with the reviewer that anthropogenic impacts of deforestation and degradation are essential processes across the tropics but argue that the model must be able to capture emergent natural dynamics first. The work of Staver et al (2011) and Flores et al (2024) demonstrate forest condition for areas of little anthropogenic disturbance, and we use this as a comparison point. We are very clear in the text that this is a potential natural emergent vegetation simulation. This is important because the model must be able to capture natural vegetation dynamics before it can be trusted to also capture anthropogenic impacts of deforestation and degradation. Anthropogenic land use inclusion in land surface models is challenging, and incorporation into a demographic model that has variable vegetation structure is more complex given the interaction between vegetation state and land use transitions. Understanding and exploring the limitations and success of natural vegetation dynamics is an essential piece of increasing our certainty in simulated vegetation dynamics that result from a combination of natural and anthropogenic disturbances and feedbacks.

4. (4) For model validation: do you use observation corresponding to the simulation period? This is not mentioned in the manuscript but I may have neglected it.

The simulation period was from 1993-2013 cycled for a period of 300 years with the final ten years used for evaluation. The GPP data was for the period from 1980-2013, LAI for the period of 2011-2015, and burned area for the period of 1997-2016. This has been added to the manuscript in section 2.1.4.

5. (5) A short paragraph in the Methods describing how forest dynamics related to disturbance are represented in FATE would be helpful for the readers to better understand the work. For lines 268-269: I don't understand whether tree mortality was simulated as partial mortality or treated as stand-replacing mortality. In the latter case, I guess, say, a 10% mortality is simulated, you just take this 10% area and start it with a new patch?

   A short paragraph explaining the three types of FATES disturbance processes (mortality of canopy trees, fire, anthropogenic disturbance) has been added to section 2.1.1 in the methods. Additional text has been added to section 2.1.2.4 to clarify that burned plants are killed and sent to coarse woody debris pools and unburned plants are added to a new patch. The newly-burned patch retains the fire-impacted vegetation structure of plants that have survived the fire event.

   The area of the burned patch is defined by the fire size, and the plant mortality relates to the intensity and duration of the fire for a combined impact according to the size and fire-tolerance and structural characteristics of the tree (height of canopy, thickness of bark, and sensitivity of vegetation canopy).

   In a situation with 10% mortality, the surviving plants would be on the newly-burned patch with a time-since-disturbance age of zero.

6. (6) The discussion is in general a little excessive given the hypothetical nature of the study. For example, lines 530-545 focus on comparing tree cover distributions with the observation but from the above we know that many reasons contribute to the disagreement; lines of 622-634 on the anthropogenic effects of fire seem not relevant with this study.

   These sections have been shortened.

Technical comments:
Line 69–70: it would be nice the authors could expand the discussions here and give a brief description of the status quo on how the effects of tree size and bark thickness on fire-caused mortality are represented, or not represented, in current fire models embedded in land surface models. I understand this needs a bit of work but it can provide a nice overview.

Thank you for this comment. Indeed, there are a small number of models that consider tree size and bark thickness. Text has been added to the introduction to highlight the variation among these models and acknowledge that some models consider tree size and some consider both tree size and bark thickness. Lines 59-66 "Some models represent fire-induced plant mortality using constant combustion and mortality factors to determine the portion of vegetation burned or killed (Rabin et al., 2017). A small set of land models represent tree mortality from fire as a function of tree size and potentially other vegetation factors. Among six land surface models that consider tree mortality from fire based on tree size, four include bark thickness as

determined by tree size as a factor, one considers bark thickness as a factor irrespective of tree size, and one does not consider bark thickness (Rabin et al., 2017). Most use the common land surface model area-averaged representation of each type of plant within a given location, which is not able to capture natural ecosystem heterogeneity or demography and potential feedback between vegetation structure and fire behavior (Fisher et al., 2018)."

Line 125: regarding the Nesterov Index. I checked that in Thonicke et al. 2010 daily maximum temperature, rather than daily temperature, was used. But in Venevsky et al. (2002), indeed daily temperature was used. So changes are needed here: (1) at least avoid citing both when using daily temperature but rather explain clear the usage in both citations and explain clearly which one was used here. (2) could you explore a bit the influence by following Venevsky rather than Thonicke et al.? This might seem minor but it would be nice to do this.

The citation to Venevsky et al 2002 has been added to the main manuscript methods in line 142, and corrected in the supplement on line 144 to clearly identify the use of Venevsky et al 2002 and not Thonicke et al 2010 for this equation. We agree that there is value in this exploration on the influence of Venevsky vs Thonicke but do not have time to address this in the revision window.

Line 113: starting from the fact that the data was used in Li et al. is not a good justification. Maybe just say that you used this dataset and Li et al. also used it? This has been modified to reflect this now.

Line 116: "under favorable conditions for burning": a crucial detail. What do you mean by 'favourable' here? How is it implemented in the model? Do you use a constant of 10% or this percentage changes with something (e.g., FDI)? This is a constant percentage, so 10% of successful lightning ignitions have the potential to cause a fire. This language has been adjusted to clarify this point.

Line 133: "Weighted averages across fuel classes": what has been used a weight? There are two options, one can either use surface-to-volume ratio or fuel weight. This is based on the fraction of each fuel as a portion of the total sum of fuels present.

Line 133-135: does FATE allow tree growing over grasses? The description here seems that CWD, dead litter and live grass form an integrated vertical profile. This is also relevant to lines 519-531. Fig. 7 shows biomass for different PFTs rather than their grid coverage and hence it is unclear whether the ground coverage of different PFTs also show a similar pattern. Trees are allowed to overgrow grasses. The grasses are considered as part of the surface fuel load for this calculation. Live trees do not contribute to the available surface fuel load.

Line 139: does 'fine' fuels mean leaf litter? Does FATE represent branches of different diameters so that it is straight forward to classify dead branches into different wood fuel classes or you need some allocation/partitioning scheme? FATES uses an allocation partitioning scheme set in the parameter file to send portions of dead biomass to CWD fraction that correspond to these fuels pools. This has been added to the text in section 2.1.2.2 at line 161-163. "A fraction of simulated biomass following tree mortality is partitioned to each of these classes as set in the parameter file (*fates_frag_cwd_frac*), which for this manuscript uses 0.045, 0.075, 0.21 and 0.67 for the 1-hr, 10-hr, 100-hr, and 1000-hr fuels respectively."

Line 145: define what is fuel moisture, because some studies use water/(water+dry fuel), some use water/(dry fuel) This has been updated in section 2.1.2.2 at line 181-183 to indicate that this includes the water and dry fuel.

Line 11&12 in Supplement: Eq xx, please check Updated reference section 3.1.2 of the supplement.
Line 13 in Supp: what do you mean by 'fuel moisture consumption'? This has been updated to indicate that it refers to fuel-specific consumption thresholds.
Eq. 3 and Eq. 4: explain what 'fc' stands for to increase readability. This has been updated to indicate that fc indicates "fuel class".
Line 166: the authors assume the reader know the meaning of 'drying raito' by default but I don't understand it. A definition for the 'drying ratio' has been added to clarify this is an empirical value used to calculate the relative fuel moisture according to a fuel type's surface area to volume.

Eq. 4: define SAV_fc This has been added. $SAV_{fc}$ is the fuel class surface area to volume ratio (cm$^{-1}$)

Line 156: $\propto$ fc, $\propto$ is a strange symbol. Is it the same sign as infinitive? Could you use something easier to understand, read and remember? What do you mean by 'user-defined'? does it mean that for every model application, e.g., in different regions, this parameter needs to be parameterized ? Or it just means that it is parmaterizable? This has been updated in section 2.1.2.2 to indicate this is the relative fuel moisture rate of drying of the fuel classes, and uses a new symbol "rel_fm$_{fc}$". The sentence has been corrected in an earlier section to indicate the *drying ratio* represents a parameterizable value used to calculate the relative fuel moisture for a particular fuel type's surface area to volume.

Lines 145-165: Overall the readability around the fuel moisture simulation part is poor. Could you improve it? More detailed descriptions will allow others to reproduce your research more easily. There was an essential fuel consumption equation missing in the main manuscript. This has been added as EQ 7 and should improve the understandability of the section. Additional text has been added to clarify parameters and their relationship to fuel moisture simulation.

Table: citation of Table 1 is poor. Some names in the first column are strange and I don't know where they are used and what they are for, e.g., Rows of 3-9. The inclusion of fuel consumption equation (EQ 7) connects these parameters to the fuel consumption methods.

Line 189-191: do you check how many fires were actually suppressed/extinguished due to this threshold? This was not tracked for this manuscript.

Line 199: l$_b$ has no unit? L$_b$ should be defined as 'major to minor axis ratio' (consistent with Eq. 14), the text gives the reverse. This has been corrected as directed.

Line 202-204: could you please detail about this error? Is it about Eq. (10) or the second half of Eq. (11)? Your second half of Eq. (11) is also different from Thonicke et al. (Eq. 13 in their articles). The correction from Wotton et al., 2009 is included in this manuscript new Eq 12 and corrects Thonicke Eq 13. Eq 13 of Thonicke adds $U_{forward}^{0.464}$ to 1.1 when it should be multiplied by 1.1. Worron et al 2009 corrects this to $(1.1*U_{forward}^{0.464})$. A note of this has been made more explciit in the manuscript in section 2.1.2.4 on line 252.

Eq. (15): a critical detail here: combing back to my comment on Line 116, is $I_{lightning}$ here exclusively scaled by 10%? Yes. For this manuscript, the value of daily lightning ignitions is the input lightning data value from the NASA LIS/OTD Gridded Climatology scaled by 10%.

Eq. (19): I found an error in the middle sub-equation when implementing spitfire in the recent ORCHIDEE trunk version. 0.563*0.22= 0.12386, which is smaller than 0.125. So there is chance that you might get negative value from this equation which is not plausible. I suggest replacing 0.125 by 0.12386 to avoid this. I found this because it prompts mass balance error in ORCHIDEE due to a negative value.

Thank you for this note. We will note this and test in further simulations.

Line 284-285: was fire module switched on from the very beginning of the simulation? I don't know if there is a risk that trees are too small at the very beginning and they get repeatedly and easily killed by fire. Do you have this issue? Yes, the fire module was on from the beginning of the simulation. In areas of frequent fire it can be challenging for trees to establish.

Fig. 2: are these results (panels a, b, c) averaged only for days with fire occurrence? How should I understand panel d? for the green line, if I accumulate the values across all months, does it mean that all grid cells have been almost twice (with accumulative value is about 2 judging by eyes)?

The results are a mean grouped by months for the final ten years of simulation. We did not restrict to only days with fire occurrence, but values of zero are not included in the average. Panel d is the average fraction burned per year in that month. For some months, such as August shown with the green line, there is recurrent burning as shown by the value greater than 1.

Fig.3: what are these temperatures? Land surface? Surface air? Fire flame? These are the 2m air temperatures. A note has been added.

Table 3: I suggest changing 'low fuel drying' to 'low fuel aridity', 'medium fuel drying' to 'medium fuel aridity'. Because there is the parameter of 'fuel drying ratio', using 'drying' in both of them makes easy confusing. This has been updated as suggested.

Table 4: Adding maximum value makes the table a lot more brain-consuming to read. Is it really necessary? Are the data mean values across all grid cells + years? The maximum value has been removed. Table caption adjusted to reflect these are for the final ten years of simulation.

Fig. 4: BeIer to use the name of fuel class (1hr, 10hr, ...) rather than 'small branch', 'twig' because the laler gives an impression that we indeed have these being represented in the model but actually we are partitioning the biomass into different fuel classes. We agree that this is an important note, and the allocation of biomass to fraction representing fuel classes has been added into the manuscript in more detail in section 2.2.2.2, and is included in table 1. For these figures we retain the use of "dead grass" and "live grass", etc to add readability and distinguish between them more easily on the figure. The differences in the fuel class parametrization for SAV, bulk density and rates of decomposition are based on measurements of samples representing these fuel classes as represented by their diameter classes, so there is an effort to tie these to measured fuel classes.

Line 373: still, from the top panels of Fig. 5, there is a tendency of higher intensity with lower live grass fuel moisture? Yes, in general there is a tendency of higher intensity with lower live grass fuel moisture.

Line 302-304: What's the role of wood density in the model or how this relates to the simulation of fire or vegetation processes? This question is also relevant for lines 467-468.

The fire-tolerant trees have a higher wood density, which in the model means they must allocate more resources to growth. With this difference in necessary resources for growth, the fire-tolerant trees have a slower growth rate than the lower wood density trees. This is detailed in section 2.1.3 lines 380-388. Additional text has been added to clarify that the fire-vulnerable tree is more likely to have crown mortality due to a higher sensitivity to leaf and crown scorch and cambial damage due to their lower bark thickness.

Line 497: 'mortality threshold' in this line implies something quite precise but I don't think there is any threshold in the model to determine whether a tree was killed completely or not. The mortality is simulated as a continuous number (fraction) indicating mortality rate? No?

You are correct. This sentence has been modified to read that the drier fuels and subsequent increase in fire intensity surpasses the fire characteristics that these fire-vulnerable trees can survive. Line 720-723 "With this shift to drier fuels and subsequent increase in fire intensity, fire associated tree mortality extends into the Amazon (Figure 8, S12) implying that the increase in fire intensity under the medium fuel drying parameterization surpasses the fire characteristics (e.g. intensity, flame height, duration) that these fire-vulnerable trees can survive."

---

## Author Comment (AC3)

**Dynamic ecosystem assembly and escaping the "fire-trap" in the tropics: Insights from FATES_15.0.0**

Jacquelyn K. Shuman, Rosie A. Fisher, Charles Koven, Ryan Knox, Lara Kueppers and Chonggang Xu
*Correspondence to*: Jacquelyn K. Shuman (Jacquelyn.k.shuman@nasa.gov)

**Response to Review**

Thank you for the comments on our manuscript. We appreciate the review and positive feedback. Specific comments are addressed in the manuscript and below with text shown in blue.

On behalf of the authors,
Jacquelyn Shuman

Huilin Huang, 19 Jan 2024

The manuscript titled "Dynamic ecosystem assembly and escaping the "fire-trap" in the tropics: Insights from FATES_15.0.0" explores the influence of climate, fire regime, and plant traits related to fire tolerance on the biogeography of tropical forests and grasslands. The authors employ simulations with the vegetation demographic model FATES-SPITFIRE, incorporating three vegetation types with varying levels of fire tolerance. The manuscript is well-written, with a clear presentation of methods, a thorough discussion of results, and a consideration of uncertainties. I have a few specific comments regarding the methodology and explanations.

L133: Please provide an explanation for $moist_{ext, fc}$, moisture of extinction

This explanation has been added to the text in section 2.1.2.2: "…the moisture content at which fuel no longer burns."

Line 325: Can you clarify the origin of the value 66,000 °C$^{-2}$? It appears to be derived from Eq. 6 in Thonicke et al. (2010) but in their paper, $1.0*10^{-3}$ is adopted.

The value of $1.0*10^{-3}$ is the drying ratio parameter used in Thonicke et al (2010). For this work we created an equation to split out the calculation for the relative rate of fuel drying ($rel\_fm_{fc}$) as it relates to the SAV (surface area to volume) of the fuel classes) and

an empirical value (Eq. 4). This allows the user to modify the parameter for the "drying ratio" without modifying the SAV of the fuel class. In Thonicke et al 2010 their Eq. 6 the values for the rate of fuel class drying are taken from the relationship between the SAV (surface area to volume) and an empirical value set to $6.6 \times 10^4$ in Thonicke et al (2010) and $1.3 \times 10^4$ in Lasslop et al 2014. This can be verified by calculating the ratio for Thonicke et al 2010 given the SAV values. For example, in this work and Thonicke et al 2010, the 10 hr small branch fuels have a SAV of 3.58 $cm^{-1}$, so using the ratio of the 3.58 $cm^{-1}$ SAV to the empirical value of $6.6 \times 10^4$ gives a drying parameter (degrees $C^{-2}$) of $5.42 \times 10^{-5}$ consistent with that reported in Thonicke et al 2010 in the section below their Eq. 6. This same calculation can be used to confirm the value for the other fuel classes; so for the 100 hr large branch fuels, SAV of 0.98 to the empirical value of $6.6 \times 10^4$ yields the value of $1.49 \times 10^5$ as reported in Thonicke et al 2010. The drying ratio value of $6.6 \times 10^4$ for Thonicke et al 2010 is also reported in Lasslop et al 2014 on page 743 in the sections above Eq. 3.

Fig. 3a shows a region of decrease in maximum temperature, whereas the surrounding regions show an increase. Can you explain the reason for the decrease?

We did not investigate this area of small decrease in maximum temperature located in the area that is dominated by a shift from trees to grasses. This region would be a good target for further investigation.

Figure 9/10 and Section 3.2: It seems the experimental design resembles sensitivity tests, as they all represent potential vegetation cases. A justification for choosing the medium drying case for validation against observations would be helpful.

Based on the histogram distribution for biomass from observations and CLM-FATES, all drying scenarios have a high bias for biomass, but the medium drying scenario captures the bimodal distribution of low and high biomass without the overprediction of low biomass found in the high drying scenario. The medium drying scenario was then used for the tropical simulation to represent this mid-range of potential vegetation for a situation without land use. The areas of high biomass within the moist tropics coincide with areas of low fire disturbance, and would expect to show high bias with the high drying scenario as well.

Fig S13. Suggest exchanging the positions of a) GPP and b) LAI to align with the color bar.

The figure was correctly laid out, but the caption was switched for the observations. This has been fixed for both S13 and S14.

Line 467-468: How does FATES determine the overcompetition of fire-vulnerable trees over fire-tolerant trees in regions where fire was absent?

In areas where fire was absent, over competition of fire-vulnerable trees would be attributed to the growth of trees in response to site and climate conditions. The fire-vulnerable trees have a lower wood density, and so are able to allocate more resources to growth than the higher wood density fire-tolerant trees. This increased growth allows the fire-vulnerable trees to out compete the fire-tolerant trees through faster biomass and height accumulation.

Line 535-536: How does FATES describe the large savanna region in Western Africa and Southern Africa, where tropical rainforest and grassland co-exist?

Similar to the results for South America, the model does not adequately capture this co-existence of trees and grasses in Africa.

---

## Author Response (AR2)

**Dynamic ecosystem assembly and escaping the "fire-trap" in the tropics: Insights from FATES_15.0.0**

Jacquelyn K. Shuman, Rosie A. Fisher, Charles Koven, Ryan Knox, Lara Kueppers and Chonggang Xu
*Correspondence to*: Jacquelyn K. Shuman (Jacquelyn.k.shuman@nasa.gov)

**Response to Review**

We thank the editor and reviewer for the careful reading and opportunity to further respond to comments. We appreciate the thorough and attentive evaluation of the manuscript, and the ability to address and clarify our response to the reviewer and within the manuscript. Specific comments are addressed in the manuscript and below with text shown in blue. Line numbers are shown for the track changes version of the manuscript.

On behalf of the authors,

Jacquelyn Shuman

Submitted on 24 Mar 2024
Referee #2:

I appreciate the author's extensive revisions on the paper. The paper could be published as is after addressing my remaining minor comments as below:

Table 2: perhaps better to make distinctions where parameters are the same between fire-vulnerable and fire-tolerant trees to make the table being easier to read. Do you really need that many digits (e.g., eaf_slatop = 0.01995827, is it really precise up to 10^-8?). Please clean a bit the digits. I hope the authors can expand their revisions to really let the readers understand Table 2. Their revisions are still too much brief for me.

Author response: We have updated the table as suggested and expanded the text explanation in methods section 2.1.3 (lines 370-372) to identify the difference between the fire-vulnerable and fire-tolerant trees more clearly. "Specifically, the fire-vulnerable and fire-tolerant trees are distinct for five parameters: leaf fire vulnerability, bark thickness, crown depth, crown mortality probability, and wood density (Table 2)."
We have added more explanation of how these parameters create a distinction between the fire-vulnerable and fire-tolerant trees in this same section preceding table 2 (lines 386-390).
"The fire-vulnerable tree has lower wood density, and from this less costly resource allocation than the fire-tolerant tree resulting in higher growth and biomass accumulation but is more likely to experience damage and mortality due to fire. The fire-vulnerable tree has higher leaf fire vulnerability, a thicker crown creating more exposure to flame scorch, a

lower accumulation of protective bark and higher probability of crown mortality than the fire-tolerant tree."

Fig. 2: are these results (panels a, b, c) averaged only for days with fire occurrence? How should I understand panel d? for the green line, if I accumulate the values across all months, does it mean that all grid cells have been almost twice (with accumulative value is about 2 judging by eyes)?: The response to this comment is rather an important detail which warrants explanation in the manuscript.

Author response: We have added detail to the manuscript results section 3.1 (lines 460-462) to explicitly identify that it was calculated for all fires and that the recurrent burning shown in Figure 2d is shown as values greater than 1.

Line 284-285: was fire module switched on from the very beginning of the simulation? I don't know if there is a risk that trees are too small at the very beginning and they get repeatedly and easily killed by fire. Do you have this issue? : The response to this comment is rather an important detail which warrants explanation in the manuscript

Author response: Thank you for this note. We have added to the manuscript that fire was active from the beginning of the simulation in both the methods in section 2.1.3 (line 365) and in the discussion section 4.1.1 (lines 756-761). As the reviewer notes, small trees are vulnerable to repeat fire. Text has been added to the discussion to further highlight that fire is present at the beginning of simulation, but that the variation in distribution across fuel drying scenarios demonstrates the importance of associated frequency of burning and fire intensity. Further study using initialization with potential stand structure would be valuable to further evaluate the 149 kW m-1 fire intensity mortality threshold derived by De Faria et al (2021) from Staver et al (2020) data and its connection to fuel moisture, fire frequency and fire intensity.
"Though fire is active from the beginning of the simulations, variation of tree distribution and biomass accumulation among the fuel drying scenarios (Figure 7) demonstrate that less frequent burning (Figure 1) and lower annual fire intensity (Figure S15) associated with wetter fuels and less fuel drying is a consideration for tree survival and distribution. Initialization with a potential tree stand structure would need to be evaluated for survival and resilience under similar fuel drying conditions and the associated fire frequency and intensity, as small stature establishing trees would be expected to show more vulnerability to fire than existing tree stands."